# CAP: Correlation-Aware Pruning
# for Highly-Accurate Sparse Vision Models

**Denis Kuznedelev**
Skoltech & Yandex
Denis.Kuznedelev@skoltech.ru

**Eldar Kurtic**
IST Austria
eldar.kurtic@ist.ac.at

**Elias Frantar**
IST Austria
elias.frantar@ist.ac.at

**Dan Alistarh**
IST Austria & Neural Magic
dan.alistarh@ist.ac.at

## Abstract

Driven by significant improvements in architectural design and training pipelines, computer vision has recently experienced dramatic progress in terms of accuracy on classic benchmarks such as ImageNet. These highly-accurate models are challenging to deploy, as they appear harder to compress using standard techniques such as pruning. We address this issue by introducing the *Correlation Aware Pruner (CAP)*, a new unstructured pruning framework which significantly pushes the compressibility limits for state-of-the-art architectures. Our method is based on two technical advancements: a new *theoretically-justified pruner*, which can handle complex weight correlations accurately and efficiently during the pruning process itself, and an *efficient finetuning procedure* for post-compression recovery. We validate our approach via extensive experiments on several modern vision models such as Vision Transformers (ViT), modern CNNs, and ViT-CNN hybrids, showing for the first time that these can be pruned to high sparsity levels (e.g. $\geq 75\%$) with low impact on accuracy ($\leq 1\%$ relative drop). Our approach is also compatible with structured pruning and quantization, and can lead to practical speedups of 1.5 to 2.4x without accuracy loss. To further showcase CAP's accuracy and scalability, we use it to show for the first time that extremely-accurate large vision models, trained via self-supervised techniques, can also be pruned to moderate sparsities, with negligible accuracy loss[1].

## 1 Introduction

Computer vision has seen impressive progress recently via a *new generation of architectures* motivated by the Vision Transformer (ViT) approach [9, 43] and its extensions, e.g. [1, 28, 46], accompanied by more advanced data augmentation and training approaches [43, 48]. Next-generation vision models such as ViT [9, 43] and ConvNext [49, 29] achieve breakthrough performance across vision tasks, despite encoding fewer inductive biases. This comes at the cost of very large computational and parameter budgets, both for training and deployment. Thus, there is a clear need to reduce these costs via *compression*, enabling deployment in resource-constrained settings.

Yet, the general consensus from the literature is that next-generation models are *harder to compress* while preserving accuracy, relative to their classic convolutional counterparts [6, 40, 35, 20]. For example, if current techniques can compress the classic ResNet50 model [18] to 80-90% unstructured sparsity with negligible loss of accuracy [33, 44], the best currently-known results for similarly-accurate ViT models only reach at most 50% sparsity while maintaining dense accuracy [6]. Our investigation of this phenomenon, detailed in the paper, shows that this occurs for two reasons:

---

[1]The code is available at https://github.com/IST-DASLab/CAP

37th Conference on Neural Information Processing Systems (NeurIPS 2023).

1. **Hardness of Pruning**: As illustrated in Figure 2, existing pruning approaches, based on magnitude [16], first-order [38] or second-order information [39] drop very significant accuracy *per pruning step* for next-generation architectures such as ViT and ConvNext.

2. **Expensive Recovery**: At the same time, recovering accuracy via fine-tuning [15] is itself computationally-challenging, as next-generation architectures are notoriously hard to train and finetune [41, 43].

Therefore, it is natural to ask whether this "lack of compressibility" is inherent for next-generation vision models, and whether their increased accuracy comes at the price of higher deployment costs.

**Contributions.** In this paper, we resolve this question by proposing a new highly-accurate *correlation-aware pruner (CAP)*, which shows that *high sparsity can be achieved across next-generation architectures* such as ViT [9, 43], ConvNext [29, 49], and augmented ResNets [48], and that this can be done in a *computationally-efficient manner*. Furthermore, CAP is compatible with other forms of compression such as token dropping and quantization, and can lead to significant inference speedups, at little or no accuracy loss.

Our primary motivation is to resolve the hardness of pruning modern, highly-accurate vision models. A key weakness of existing state-of-the-art approaches [10, 38, 6, 33, 39, 14], is that they *do not directly take into account weight correlations:* At each pruning step, a *saliency score* is computed per weight, e.g., the magnitude, weights are ranked by this score, and a large subset is chosen to be removed and/or re-introduced. However, this does not take into account the fact that removed weights *may themselves be correlated*: for example, a pair of linearly-dependent rows in a layer's weight matrix may seem *individually easy to remove*, since they are mutually-redundant, but removing *both* at a step may lead to a large accuracy drop. This phenomenon is especially-relevant for Transformer-based architectures, which encode more complex inter-weight correlations, relative to their convolutional counterparts, in which correlations tend to be localized [39].

We circumvent this issue via a novel formulation of the constrained optimization problem corresponding to choosing the optimal weights to prune at a step [17], while taking correlations into account. We prove formally that this task can be reduced to finding the set of sparse weights which best preserve the correlation between the dense weights and their gradients, on a given set of training samples. This allows us to efficiently solve the "optimal pruning with correlations" problem, via a fast approximate solver for the above constrained optimization problem.

The CAP algorithm outperforms all other known pruners by a considerable margin (see Figure 6). In turn, this precision lends greater flexibility in *gradual pruning*, and enables us to build a *computationally-efficient* approach to compress next-generation vision models, addressing our second motivating challenge. Specifically, our gradual pruning approach provides a simple and general recipe combining data-efficient augmentation and regularization [43, 41] with theoretically-justified learning rate rewinding [36, 23], leading to state-of-the-art sparsity-vs-accuracy trade-offs.

For instance, experiments on the standard ImageNet-1K benchmark [37] show for the first time that ViT models can attain high sparsity levels without significant accuracy impact: specifically, we can achieve 75-80% sparsity with relatively minor ($< 1\%$) accuracy loss, and 90% sparsity with moderate loss. In turn, this sparsity leads to computational speedups of more than 2x. Our approach extends to highly-accurate pruning of large ViTs trained by self-supervised pretraining, but also to other modern models, such as ConvNext [29, 49] or highly-accurate ResNets [48].

In sum, our results show that next-generation highly-accurate vision architectures are still highly-compressible, but may also require next-generation pruning approaches.

**Related work.** Next-generation vision architectures such as Vision Transformers (ViTs) [9] and ConvNext [29, 49] have set new accuracy benchmarks, but are known to require careful tuning in terms of both augmentation and training hyper-parameters. Identifying efficient recipes is an active research topic in itself [43, 41]. We propose simple and general recipes for *fine-tuning* such models, which should be useful to the community. Several prior works have investigated ViT compression, but focus on *structured* pruning, such as removing tokens [54, 22, 50, 31, 40, 35, 20]. Our experiments show that structured approaches are *orthogonal* to CAP, which can be applied in conjunction to structured compression, to obtain further gains.

The only existing prior work on *unstructured* ViT pruning is SViTE [6], which performed careful customization of the RigL pruning method [10] to the special case of ViT models. We also present

results relative to other methods, such as tuned magnitude pruning, the best first-order and second-order pruners [39, 38, 14, 23] and AC/DC pruning [33]. (These methods are known to outperform all other prior methods.) CAP improves upon existing methods across almost all benchmarks, by large margins at high sparsity.

Research on accurate pruning using second-order information was initiated by LeCun et al. [26], and has recently garnered significant attention [8, 45, 39, 52]. This approach can lead to good results for both gradual pruning [39] and one-shot (post-training) compression [12]. Existing such pruners are not correlation-aware, and are outperformed by CAP across all of our experiments.

## 2 Background and Problem Setup

The pruning problem assumes a fixed model architecture with weights $\mathbf{w} \in \mathbb{R}^d$ ($d$ is the total number of parameters), and aims to find a configuration of weights with as many zeros as possible while preserving the performance of the original dense model. *Gradual* pruning, e.g. [19], usually starts from an accurate *dense* model, and progressively removes weights by setting them to zero, followed by fine-tuning phases.

**Weight Saliency.** The pruning step usually relies on proxies for weight importance, defined according to certain criteria. For instance, the *weight magnitude* is arguably the most popular criterion, e.g. [16, 53, 15]. Specifically, given model $\mathbf{w} \in \mathbb{R}^d$, the saliency of each weight is its absolute value (the magnitude) $\rho_j = |w_j|$ for $j \in \{1, 2, \ldots, d\}$; weights with the smallest scores are pruned away. Gradual magnitude pruning is usually a strong baseline across most models and settings. Many other criteria exist, such as gradient magnitude [10] or "rates of change" in the weights [38].

**The Optimal Brain Surgeon (OBS).** LeCun et al. [26] and Hassibi et al. [17] obtained weight saliency scores by leveraging (approximate) second-order information about the loss, starting from the Taylor approximation of the loss $\mathcal{L}$ in the vicinity of the dense model parameters $\mathbf{w}^*$. Assuming that $\mathbf{w}^*$ is close to the optimum (hence $\nabla \mathcal{L}(\mathbf{w}^*) \simeq 0$), one seeks a binary mask $\mathbf{M}$ (with elements $\in \{0, 1\}$) and new values for the remaining weights $\mathbf{w}^M$, such that the resulting increase in loss is minimal. A standard approach to approximate the loss increase is to expand the loss function up to the second order in model weights:

$$\mathcal{L}(\mathbf{w}^M) - \mathcal{L}(\mathbf{w}^*) \simeq \frac{1}{2}(\mathbf{w}^M - \mathbf{w}^*)^\top \mathbf{H}_\mathcal{L}(\mathbf{w}^*)(\mathbf{w}^M - \mathbf{w}^*) \tag{1}$$

where $\mathbf{H}_\mathcal{L}(\mathbf{w}^*)$ is the Hessian of the model at $\mathbf{w}^*$, and $\mathbf{w}^M$ represents weights after the pruning step. In this setup, [26] and [17] showed that the "optimal" weight to remove, incurring the least loss, and the update to the remaining weights, can be determined via a *closed-form* solution to the above inverse problem. Specifically, the saliency score $\rho_i$ for $i^{\text{th}}$ weight and the optimal weight update $\delta \mathbf{w}$ for the remaining weights after elimination of the $i^{\text{th}}$ weight are as follows:

$$\rho_i = \frac{w_i^2}{2[\mathbf{H}_\mathcal{L}^{-1}(\mathbf{w}^*)]_{ii}} \quad \text{and} \quad \delta \mathbf{w}^* = -\frac{w_i}{[\mathbf{H}_\mathcal{L}^{-1}(\mathbf{w}^*)]_{ii}} \mathbf{H}_\mathcal{L}^{-1}(\mathbf{w}^*)\mathbf{e}_i, \tag{2}$$

where $\mathbf{e}_i$ is the $i^{\text{th}}$ basis vector. Theoretically, the procedure would have to be executed one-weight-at-a-time, recomputing the Hessian after each step. In practice, this procedure suffers from a number of practical constraints. The first is that direct Hessian-inverse computation is computationally-infeasible for modern DNNs, due to its quadratic-in-dimension storage and computational costs. This has led to significant recent work on efficient second-order approximations for pruning and quantization [8, 45, 52].

**WoodFisher and the Optimal BERT Surgeon.** The *empirical Fisher* approximation [2] is a classic way of side-stepping some of the above constraints, and can be formally-stated as follows:

$$\mathbf{H}_\mathcal{L}(\mathbf{w}^*) \simeq \mathbf{F}(\mathbf{w}^*) = \lambda \mathbf{I}_{d \times d} + \frac{1}{N} \sum_{i=1}^{N} \nabla \mathcal{L}_i(\mathbf{w}^*) \nabla \mathcal{L}_i(\mathbf{w}^*)^\top \tag{3}$$

where $\nabla \mathcal{L}_i(\mathbf{w}^*) \in \mathbb{R}^d$ is a gradient computed on a sample of data, $\lambda > 0$ is a dampening constant needed for stability, and $N$ is the total number of gradients used for approximation. Note that the resulting matrix is *positive* definite.

The memory required to store the empirical Fisher matrix is still quadratic in $d$, the number of parameters. Singh and Alistarh [39] investigated a diagonal block-wise approximation with a predefined block size $B$, which reduces storage cost from $\mathcal{O}(d^2)$ to $\mathcal{O}(Bd)$, and showed that this approach can lead to strong results when pruning CNNs. Kurtic et al. [23] proposed a formula for pruning fixed groups/patterns (e.g., 4 consecutive weights), together with a set of non-trivial optimizations to efficiently compute the Fisher block inverses, which allowed them to scale the approach for the first time to large language models.

A second obvious limitation of the OBS framework is that applying the procedure and recomputing the Hessian one weight at a time is prohibitively expensive, so one usually prunes multiple weights at once. Assuming we are searching for the set of weights $Q$ whose removal would lead to minimal loss increase after pruning, we get the following constrained optimization problem:

$$\min_{\delta\mathbf{w}} \frac{1}{2}\delta\mathbf{w}^\top \mathbf{F}\left(\mathbf{w}^*\right)\delta\mathbf{w} \quad \text{s.t.} \quad \mathbf{E}_Q\delta\mathbf{w} + \mathbf{E}_Q\mathbf{w}^* = \mathbf{0}, \tag{4}$$

where $\mathbf{E}_Q \in \mathbb{R}^{|Q|\times d}$ is a matrix of basis vectors for each weight in $Q$. The corresponding saliency score for the group of weights $Q$ and the update $\delta\mathbf{w}_\mathbf{Q}^*$ of remaining weights are [23]:

$$\rho_Q = \frac{1}{2}\mathbf{w}_Q^{*\top} \left(\mathbf{F}^{-1}(\mathbf{w}^*)_{[Q,Q]}\right)^{-1} \mathbf{w}_Q^* \text{ and } \delta\mathbf{w}_Q^* = -\mathbf{F}^{-1}(\mathbf{w}^*)\mathbf{E}_Q^\top \left(\mathbf{F}^{-1}(\mathbf{w}^*)_{[Q,Q]}\right)^{-1} \mathbf{w}_Q^*. \tag{5}$$

However, an exhaustive search over all subsets of size $|Q|$ from $d$ elements requires $\binom{d}{|Q|}$ evaluations, which makes it prohibitively expensive for $|Q| > 1$. For unstructured pruning, virtually all known techniques, e.g. [39, 14], ignore correlations between weights. Similarly, for group pruning, Kurtic et al. [23] ignore correlations between groups. Despite these approximations, both approaches yield state-of-the-art results in their respective setups. As we will demonstrate later, our CAP method improves upon these approximations by reformulating this problem and proposing a correlation-aware solution that is fast and memory-efficient even for models with $\sim 100\text{M}$ parameters.

## 3 The CAP Pruning Framework

We introduce new techniques to address both the hardness of pruning modern vision architectures, and their high computational cost for fine-tuning: we introduce a new state-of-the-art one-shot pruner, which is complemented with a simple and general framework for data-efficient fine-tuning.

### 3.1 Ingredient 1: Efficient Correlation-Aware Pruning

Our aim is to solve the pruning problem stated in the previous section: given a weight pruning target $k$, find the optimal set of weights $Q$ to be pruned, such that $|Q| = k$ and the loss increase is minimized. Exactly solving for the optimal $Q$ is an NP-hard problem [3], so we will investigate an iterative greedy method for selecting these weights, similar to the ideal version of the OBS framework discussed above. Importantly, our method *properly considers weight correlations*, which turn out to be important since, as demonstrated in Appendix Q, the empirical Fisher has an apparent non-diagonal structure, while being *fast and space-efficient*. In turn, this leads to significant improvements over other pruners, especially in the context of vision transformers.

Formally, a correlation-aware greedy weight selection approach would perform pruning steps iteratively, as follows. Given a set of already-selected weights $Q$, initially $\emptyset$, we always add to $Q$ the weight $q$ which has minimal joint saliency $\rho_{Q\cup\{q\}}$, repeating until the size of $Q$ equals the pruning target $k$. The fact that we add weights to the set one-by-one allows us to take into account correlations between pruned weights. However, a naive implementation of this scheme, which simply recomputes saliency at each step, would be prohibitively expensive, since it requires $O(kd)$ evaluations of $\rho_Q$, each of which involves an $O(B^3)$ matrix inversion, where $B$ is the Fisher block size.

**Disentangling Correlations.** The centerpiece of our approach is a reformulation of the OBS multi-weight pruning problem in Equation 5 which will allow us to take correlations into account, while being practically-efficient. Specifically, we now show that, when using the empirical Fisher approximation, the problem of finding the optimal set of weights $Q$ to be removed, while taking correlations into account, is equivalent to the problem of finding the set of sparse weights which best preserve the original correlation between the dense weights $\mathbf{w}^*$ and the gradients $\nabla\mathcal{L}_i(\mathbf{w}^*)$ on an

fixed set of samples $i \in \mathcal{S}$. Formally, this result, whose proof we provide in Appendix O, is stated as follows.

**Theorem 3.1.** *Let $\mathcal{S}$ be a set with $m$ samples, and let $\nabla \mathcal{L}_1(\mathbf{w}^*), \dots, \nabla \mathcal{L}_m(\mathbf{w}^*)$ be a set of their gradients, with corresponding inverse empirical Fisher matrix $\mathbf{F}^{-1}(\mathbf{w}^*)$. Assume a sparsification target of $k$ weights from $\mathbf{w}^*$. Then, a sparse minimizer for the constrained squared error problem*

$$\min_{\mathbf{w}'} \frac{1}{2m} \sum_{i=1}^{m} \left( \nabla \mathcal{L}_i(\mathbf{w}^*)^\top \mathbf{w}' - \nabla \mathcal{L}_i(\mathbf{w}^*)^\top \mathbf{w}^* \right)^2 \tag{6}$$

*s.t. $\mathbf{w}'$ has at least $k$ zeros, is also a solution to the problem of minimizing the Fisher-based group-OBS metric*

$$\arg\min_{Q, |Q|=k} \frac{1}{2} \mathbf{w}_Q^{*\top} \left( \mathbf{F}^{-1}(\mathbf{w}^*)_{[Q,Q]} \right)^{-1} \mathbf{w}_Q^*. \tag{7}$$

**A Fast Solver.** The formulation in Equation (6) reduces pruning to a sparse regression problem, where the "input" is given by *gradients* over calibration samples. A related problem arises in the context of one-shot (post-training) pruning [21, 12], where the authors solve a sparse $\ell_2$-fitting problem, but sparse weights are determined relative to the *layer inputs* rather than the *layer gradients*. Specifically, the OBC solver [12] utilizes that, due to the quadratic loss, the per-row Hessians are independent of both the weights and each other. Thus the solver processes matrix rows sequentially, greedily eliminating weights, one-by-one, in increasing order of squared error while always updating all remaining weights to compensate for weight removal as much as possible. This essentially implements the OBS selection and update in Equation 2 *exactly*, assuming layer-wise $\ell_2$ loss. We extend this strategy to efficiently implement our new Fisher-based greedy weight-subset selection.

A direct application of the OBC approach to remove $\Theta(d)$ weights would have $O(d^3)$ runtime, where $d$ is the layer dimension, as the $\Theta(d^2)$-time selection + update process is repeated $\Theta(d)$ times. By utilizing the block-diagonal structure of the Fisher matrix with block size $B$ during the update after each weight elimination, this complexity can be reduced to $O(d \cdot \max(d, B^2))$, which is however still too slow as $B$ is typically much smaller than $d$. Instead, we proceed differently: we apply the sparse regression solver to each individual Fisher block separately and globally merge results only in the very end. This allows us to efficiently simulate global weight saliency calculation and weight updates following Equations 5 and 7, in a block-Fisher formulation. The full algorithm requires $O(d \cdot B^2)$ runtime and $O(d \cdot B)$ space and is detailed in Appendix 1—-an efficient implementation is also provided in the Supplementary.

The key difference of our approach relative to WoodFisher is that updates are continuously executed during subset-selection and we are thus explicitly considering the correlations captured by the off-diagonal Fisher elements. When working with small block sizes, our method is very fast and has practically no overhead over existing Fisher-based OBS approaches, while yielding significantly improved one-shot pruning results (see e.g. Figure 2 and the accompanying discussion).

**An Faster Variant for Massive Models.** In Appendix B, we present a version of CAP which scales efficiently to models with more than 1 billion parameters, by leveraging additional approximations in the Fisher matrix structure and by relaxing the pruning order.

### 3.2  Ingredient 2: Fine-tuning and Pruning Procedure

To achieve best performance, modern training procedures involve longer training schedules together with a careful choice of hyperparameters, since these are known to have a major impact on convergence and accuracy [43, 41, 48]. We found the same to apply to post-pruning accuracy recovery, which is key in gradual pruning; below, we describe the main ingredients to obtaining highly-accurate fine-tuning schedules as part of our method.

**Learning Rate Schedule.** First, to achieve good performance during gradual pruning, the learning rate (LR) schedule is crucial. Specifically, we propose to use a *cyclic linear* schedule:

$$\eta(t) = \eta_{\max} - (\eta_{\max} - \eta_{\min}) \frac{t \% T}{T}, \tag{8}$$

where $\% x$ means taking the remainder after integer division by $x$. We chose a linear decay for simplicity; we obtained similar results for other functions (e.g., cubic decay). By contrast, as we

illustrate in Figure 1, the *cyclic* nature of the schedule is key for accurate pruning, as well as for efficient sparsity sweeps (see below).

**Theoretical Justification.** Specifically, this choice is justified theoretically by tying back to the original assumptions of the OBS framework: for Equation 1 to hold, the pruned model should be well-optimized (i.e. have small gradients) at the point when pruning is performed. Moreover, right after the pruning step, having a larger value of the learning rate is useful since it gives the model a chance to recover from the sub-optimal point induced via pruning. We note that this learning rate schedule is different from prior work on pruning, which typically uses a single decay cycle [25, 39, 33], or dynamic learning rate rewinding, e.g. [11, 36].

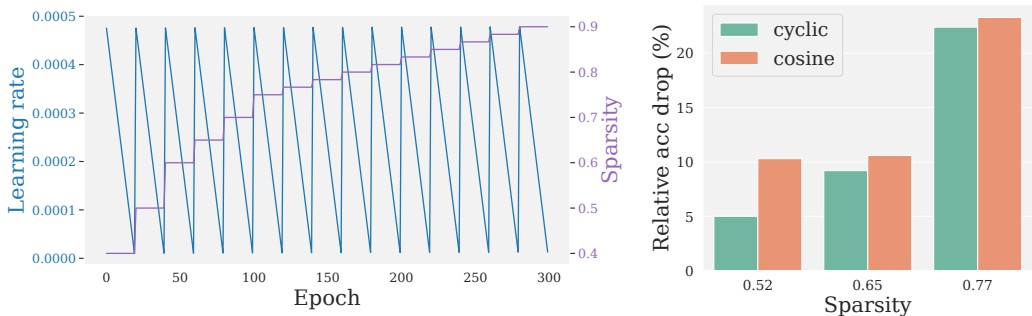

Figure 1: (**left**): Blue: Cyclic linear learning rate schedule used in the work. Violet: Dependence of the global model sparsity on the epoch. Every change in sparsity corresponds to a pruning step. (**right**): Relative accuracy drop (i.e difference between validation accuracy before and after pruning update) for training with *cyclic* and *cosine* schedule, respectively.

**Regularization and Augmentation.** For augmentation, similarly to [6], we adopt known best-practices from the literature: smaller models such as DeiT-Tiny benefit from *lower* levels of data augmentation during fine-tuning as in [41], whereas larger models such as DeiT-Base behave best with more complex augmentation and regularization [43]. This is intuitive, since fine-tuning sparsified small models with high augmentation may exceed model capacity, rendering the optimization process unstable. We provide detailed parameter values and ablations in Appendix C.

**Augmentation for the Empirical Fisher.** The choice of augmentation is of great importance not only for the model training but for the accurate estimate of the Empirical Fisher as well. We observed that without proper augmentation the sparse solution obtained overfits to the training data even when finetuned with the same augmentation and regularization setup. We provide details in Appendix P.

**Efficient Sparsity Sweeps.** We propose a simple iterative pruning framework, which takes a set of target sparsity configurations and produces models which match these configurations *in a single run*. Specifically, we start from a standard gradual pruning setup, which prunes in a sequence of steps of increasing sparsity, followed by sparse fine-tuning. We then set the intermediate values in such a way that all intermediate target sparsity levels are achieved. For example, if one wishes to obtain checkpoints with sparsity levels $40\%, 50\%, 75\%, 90\%$, one can set the lowest sparsity level on the gradual pruning schedule to $40\%$, the highest sparsity level to $90\%$, and $50\%, 75\%$ as intermediate points. Between any two such pruning steps, we apply the cyclic retraining schedule above, which ensures that all intermediate points are sufficiently optimized.

We emphasize that *the accurate CAP pruner is key* for efficient pruning: virtually all previous high-accuracy pruning methods in this setting, e.g. [25, 39, 6] redo the *entire training run* for each sparsity target. In our experiments, we also examine the impact of additional fine-tuning applied to each checkpoint, and show that it induces small-but-consistent improvements.

## 4 Experimental Setup and Results

**Setup and Goals.** We consider the ImageNet [37] image classification benchmark, and aim to examine how sparsity impacts accuracy for different model variants. We consider three scenarios: *one-shot, single-step pruning* of a pretrained model, where performance is clearly tied to the quality of the second-order approximation, *one-shot + fine-tuning*, in which we follow one-shot pruning

by a short period of fine-tuning, and, finally, *iterative gradual pruning*, where one applies pruning periodically, with some retraining interval, gradually increasing sparsity.

## 4.1 One-shot pruning, without and with fine-tuning

We start by examining the quality of existing one-shot pruners relative to CAP. We compare against carefully-tuned variants of Magnitude Pruning (Magn), First-Order Gradient times Weight (GrW) [38], and the SOTA second-order methods WoodFisher [39, 23] and M-FAC [14]. Our tuning process, optimal hyper-parameter choices, and ablations are detailed in Appendices J and K. WF-1 represents the *diagonal approximation* of the Hessian proposed by the original OBD [26], scaled via the WoodFisher implementation. For Magnitude and CAP we present results for both *uniform* layer-wise sparsity and *global* sparsity. For all other methods, we present results for global sparsity, which yields better results, as the methods can adjust sparsity globally. We only investigate global sparsity for the other methods in Figure 2.

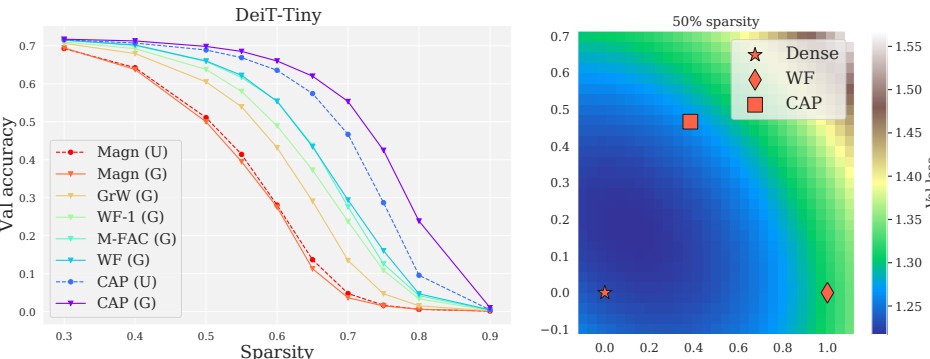

Figure 2: One-shot pruning for DeiT-Tiny.    Figure 3:  Validation loss surface.

The full comparison is presented in Figure 2 for DeiT-Tiny. Notice that all first and second-order methods outperform Magnitude, and that all 2nd-order methods are better than the 1st order saliency. WoodFisher with small block size is better than both the diagonal approximation and large-block M-FAC. This suggests that it is beneficial to take weight correlations into account, but attempting to incorporate dependencies between large groups of weights may lead to noisy estimates detrimental to performance. We also present a comparison on other models (ViT and ResNet) in Appendix E.

CAP outperforms all other methods, by a large margin. Remarkably, the gap is so large that *CAP with uniform sparsity* still outperforms WoodFisher with *global sparsity*, which can re-distribute sparsity across layers. The computational cost of WF is approximately the same as for CAP: CAP pruning step on DeiT-Small takes 23 minutes, compared to 20 minutes for WF. (The majority of the cost for both methods comes from the collection of gradients, not Hessian estimation.)

To investigate the performance difference between CAP and WF, we pruned several variants of ViT models to 50% and compared the mask overlap between WF and CAP via their IoU (intersection over union). We observe that the sparsification masks differ significantly: the IoU between the WF and CAP mask is 82.7%, 80.5%, 73.3% for DeiT-Tiny, DeiT-Small, and DeiT-Base, respectively, suggesting that the weight correlations taken into account by CAP lead to significantly different pruning decisions. The same trend is visible in the validation loss surface, projected on the plane in weight space, for the dense model and the 50% sparse models via WF and CAP. (We adopt the approach from [32] for loss surface visualization. Specifically, the unit vector in the horizontal direction is chosen to be $w_{WF} - w^*$ and the vertical direction is defined by the component of $w_{CAP} - w^*$ orthogonal to the horizontal axis. Above $w^*, w_{WF}, w_{CAP}$ are the weights of the dense model and sparse solutions found by WF and CAP, respectively.) Figure 3 shows that CAP chooses a significantly less steep direction in the loss basin compared to WF.

**Compressing highly-accurate models.** Modern pretraining methods [7, 5, 34, 51] in conjunction with large vision backbones achieve extremely high accuracy on standard benchmarks such as ImageNet-1K. We leverage the scalability and accuracy of CAP to investigate, for the first time, sparsity in such highly-accurate models, in the absence of finetuning. Specifically, we start from the

ConvNext-Large checkpoint pretrained via CLIP on the LAION-2B dataset and finetuned sequentially on ImageNet-21k and ImageNet-1k [47].

Table 1: Accuracies of CLIP-pretrained ConvNext-L on ImageNet-1k, following one-shot pruning.

| Model | Method | Sparsity (%) | Top1-Accuracy (%) |
|---|---|---|---|
| | Dense | 0 | 87.8 |
| | GM | | 80.2 |
| | WF | 50 | 86.6 |
| | CAP | | **87.5** |
| | GM | | 38.7 |
| ConvNext-L | WF | 60 | 85.1 |
| | CAP | | **87.1** |
| | GM | | 0.5 |
| | WF | 70 | 73.7 |
| | CAP | | **86.8** |

In Table 1, we compare our method with WoodFisher (WF) and Global Magnitude (GM) for one-shot pruning via several sparsity targets. Results show that 1) CAP can induce 50% sparsity with relatively low (0.3%) accuracy loss, and 60% with less than 1% accuracy drop; 2) CAP significantly outperforms other methods. In addition, we find that these results can be improved with a limited amount of fine-tuning; please see Appendix E for full results.

## 4.2 Gradual Pruning Results

Finally, we execute gradual pruning with the sparsity schedule, augmentation choices, and cyclic linear learning-rate scheduler discussed above. The whole gradual pruning procedure lasts for 300 epochs, as in [43]. We aim to obtain accurate sparse checkpoints for 50%, 60%, 75%, 80%, and 90% sparsity. For this, we prune to 40% in the initial step, and increment sparsity every 20 epochs, until reaching 90%, with fine-tuning in between. (See Appendix J for full results and ablations.) We select the accuracy of intermediate models which match the target sparsities; to examine the impact of fine-tuning, we trained each of the resulting sparse checkpoints for an additional 100 epochs, marked with (↑). We compare with global magnitude (GM) following the same schedule as CAP, as well as the state-of-the-art SViTE [6] paper, which trains the sparse model from scratch using a variant of RigL [10], but for a total of 600 epochs. The results are in Table 2.

Table 2: Results for gradual pruning of DeiT-Tiny and Small models on ImageNet-1k. CAP achieves 50% sparsity without accuracy loss, and 80% sparsity with less than 1% relative error.

| Model | Method | Sparsity (%) | FLOP Reduction (%) | Top1 Accuracy (%) | Model | Method | Sparsity (%) | FLOP Reduction (%) | Top1 Accuracy (%) |
|---|---|---|---|---|---|---|---|---|---|
| | Dense | 0 | 0 | 72.2 | | Dense | 0 | 0 | 79.8 |
| | GM | | 43.9 | 73.5 | | GM | | 46.7 | 79.3 (79.8 ↑) |
| | CAP | 50 | 43.9 | **73.7** | | CAP | 50 | 46.9 | 79.4 (**79.9** ↑) |
| | SViTE-Tiny | | 43.9 | 69.6 | | SViTE-Small | | 46.3 | 79.7 |
| | GM | 60 | 52.6 | 73.1 (73.2 ↑) | | GM | 60 | 56.1 | 79.0 (79.5 ↑) |
| | CAP | | 52.6 | 73.3 (**73.6** ↑) | | CAP | | 56.2 | 79.3 (**79.8** ↑) |
| DeiT-Tiny | | | | | | SViTE-Small | | 55.4 | 79.4 |
| | GM | | 65.8 | 71.4 (71.9 ↑) | DeiT-Small | GM | | 70.1 | 78.0 (78.7 ↑)) |
| | CAP | 75 | 65.8 | **72.3** (**72.6** ↑) | | CAP | 75 | 70.2 | 78.5 (**79.0** ↑) |
| | SViTE-Tiny | | 65.8 | 63.9 | | SViTE-Small | | 70.3 | 77.0 |
| | GM | 80 | 69.7 | 70.5 (70.9 ↑) | | GM | 80 | 74.2 | 77.3 (77.9 ↑) |
| | CAP | | 70.2 | 71.7 (**72.0** ↑) | | CAP | | 74.9 | 78.0 (**78.6** ↑) |
| | GM | | 79.0 | 66.2 (66.6 ↑) | | GM | | 84.0 | 74.1 (74.7 ↑) |
| | CAP | 90 | 79.0 | 67.4 (**68.0** ↑) | | CAP | 90 | 84.1 | 75.2 (**75.8** ↑) |
| | SViTE-Tiny | | 79.0 | 49.7 | | SViTE-Small | | 84.1 | 70.1 |

For DeiT-Tiny and 50% sparsity, we achieve significant improvements upon SViTe, and even manage to improve test accuracy relative to the dense model. We believe this is due to the choice of augmentation during fine-tuning, and possibly due to regularizing effects of sparsity. At 75-80%, we recover the dense model accuracy. We observe a significant accuracy drop only at 90%. GM pruning also benefits from the choices made in our schedule, outperforming SViTE at 50% sparsity; yet, there are significant gaps in favor of CAP at higher sparsities, as expected.

On the 4x larger DeiT-Small model, SViTE performs remarkably well at 50% sparsity (79.7%), almost matching the dense model, but CAP outperforms it slightly after fine-tuning (79.9%). In terms

of total training budget, SViTE uses 600 epochs to produce each model (and so, the 50%-sparse one as well), whereas we use a total of 40 epochs for gradual pruning to 50% + initial fine-tuning, and 100 additional epochs for sparse model fine-tuning. Even if we take into account the original 300 epochs for training the publicly-available dense DeiT checkpoint [43], our approach is significantly more efficient (440 vs. 600 epochs). The cost savings compound across sparse models, since we obtain all our pruned models from the same end-to-end gradual run. At 75% sparsity, CAP drops $\sim 1\%$ of accuracy relative to dense post-finetuning, with a significant gap of $1\%$ Top-1 relative to GM, and $2\%$ Top-1 relative to SViTE. The trend continues for higher sparsities, where we note a remarkable gap of $5.7\%$ Top-1 vs SViTE at 90% sparsity. We obtain similar numbers for DeiT-Base in Table 3; generally, we achieve $\geq 99\%$ recovery at $\geq 75\%$ sparsity, showing for the first time that ViT models can be pruned to such sparsities with marginal accuracy loss.

| Model | Method | Sparsity (%) | FLOP Reduction (%) | Top1 Accuracy (%) |
|---|---|---|---|---|
| | Dense | 0 | 0 | 81.8 |
| | CAP | 50 | 48.5 | **81.6** |
| | SViTE-Base | | 48.0 | 81.5 |
| DeiT-Base | CAP | 60 | 58.2 | **81.5** |
| | SViTE-Base | | 57.5 | 81.3 |
| | | 75 | 72.8 | 81.1 (81.2 ↑) |
| | CAP | 80 | 77.7 | 80.8 (81.1 ↑) |
| | | 90 | 87.4 | 79.7 (80.1 ↑) |

Table 3: Accuracy results for gradual pruning of DeiT-Base model on ImageNet-1k.

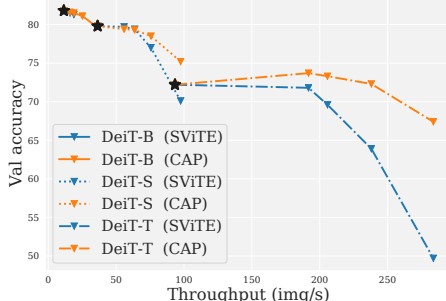

Figure 4: Accuracy vs. real-time throughput for dense ($\star$) and sparse ViT-models.

**Sparse Speedups.** The results in Figure 4 examined the speedups obtained by CAP from unstructured sparsity for {50%, 75%, 90%}-sparse ViT-family (base, small, tiny) models, when executed on a sparsity-aware CPU inference engine [24]. Specifically, we executed the models from Table 2 using 4 cores of an Intel(R) Xeon(R) Gold 6238R CPU, at batch size 64. We find it interesting that sparse ViTs build an almost-contiguous Pareto frontier from 82% to 68% Top-1 accuracy (Y axis), with a 25x span in throughput (from 11 imgs/second to 260 imgs/second, X axis). Notably, the DeiT-Tiny model obtains a speedup of 2.4x without any accuracy loss, while Base and Small ViTs show 1.5x speedups with minimal loss of accuracy. Thus, these results show that unstructured sparsity can be a very promising approach to speeding up ViTs. We note that competing methods (e.g. SViTE) would provide similar speedups at the same sparsity levels, but significantly lower accuracy, as shown in the figure, as well as in Table 2.

**Additional Results and Details.** Due to space constraints, full ablations and several additional experiments have been deferred to the Appendix. We present the full pseudocode for CAP in Appendix A, and a faster approximate version of CAP in Appendix B. We describe all parameters and perform additional ablations in Appendices C and J. We present results for one-shot pruning across several other models, as well as with short amounts of fine-tuning, in Appendix E. We present gradual pruning results for several other vision models (EfficientFormer, ResNet50D, and EfficientNetV2) in Appendix F and demonstrate that one can compress them to high sparsity as well, using our approach. In Appendix H we show that CAP comes with little additional computational cost compared to WoodFisher [23]. In Appendix I we show that CAP can also be applied in conjunction with other types of compression, in particular token pruning [35], quantization-aware training, and semi-structured (2:4) sparsity. In Appendix M, we show that CAP also outperforms AC/DC pruning [33], whereas Appendix N contains results for the DeTR detection model. In Appendix G we study the scaling behavior of sparsity solvers with respect to the ConvNext2 model family.

## 5 Discussion

We presented a new correlation-aware pruner called CAP, which sets a new state-of-the-art sparsity-accuracy trade-off. We have shown for the first time that ViT variants can support significant weight pruning ($\geq 75\%$) at relatively minor accuracy loss ($\leq 1\%$), inducing compression trade-offs that similar to those of CNNs, and that CLIP-pretrained, highly-accurate models can also support sparsity accurately. Thus, despite weaker inductive biases, next-generation vision models do not require over-parametrization, and can be competitive with CNNs in terms of accuracy-per-parameter. Our approach is complementary to other compression approaches, leading to significant speedups.

## 6 Acknowledgements

This project has received funding from the European Research Council (ERC) under the European Union's Horizon 2020 research and innovation programme (grant agreement No 805223 ScaleML). The authors would also like to acknowledge computational support from the ISTA IT department, in particular Stefano Elefante, Andrei Hornoiu, and Alois Schloegl, as well as Amazon EC2 for research credits. D.K. was supported by Russian Science Foundation, grant 21-11-00373.

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

# A    CAP algorithm description

The section below illustrates in the CAP pruning algorithm step-by-step. Prunable model weights $\mathbb{R}^d$ are partitioned into blocks of fixed size $B$. Below $\rho_i^{(B)}$ denotes the saliency scores for weight $i^{\text{th}}$ inside a block it belongs to and $\rho_i$ is the score across the whole model. The steps of the algorithm are listed below:

---

**Algorithm 1** CAP pruning algorithm

---

1: $\rho_i$ - saliency scores for weights
2: Accumulate Fisher inverse blocks $\mathbf{F}$
3: **for** each block **do**
4:     err $= 0$
5:     **for** element in a block **do**
6:         Select the weight $w_i$ with smallest score $\rho_i^{(B)}$ (using the (2) for $\rho_i$)
7:         Prune $w_i$
8:         Update remaining weights in the block via (2)
9:         err$+ = \rho_i^{(B)}$
10:         $\rho_i \leftarrow$ err
11:         Save current state of the block for later merging
12:         Update Fisher inverse block
13:     **end for**
14: **end for**
15: Sort the scores $\rho_i$ in ascending order
16: Mark the weights with smallest scores $\rho_i$ as pruned
17: **for** each block **do**
18:     Load the saved state of the block with the weights marked pruned and all remaining alive.
19: **end for**

---

# B    FastCAP: A faster version of the algorithm

CAP can scale to models with < 1B parameters, covering most standard vision models. Its key computational and memory cost comes because of computing and maintaining the inverse Fisher approximation, similarly to WoodFisher/Optimal BERT Surgeon.

To address the scalability issue we present a much more scalable approximate variant of CAP, called FastCAP, which is inspired by the recent SparseGPT scalable pruner [13]. Specifically, FastCAP reduces compute and memory cost by $\sim$ 100x by leveraging a cheaper Fisher approximation and relaxing the optimal greedy order of pruning. FastCAP can be scaled to billion-parameter vision models, i.e. we can prune ViT-Giant in a few minutes, while achieving reasonable performance in one shot-setting for 50-70% sparsity.

The main two components of the CAP/FastCAP algorithm are:

- **Empirical Fisher estimate.** The original CAP has specific Fisher block for each output channel, whereas the FastCAP version averages blocks across the output channel dimension. This is a further approximation, but significantly reduces storage complexity.

- **Weight elimination and update iteration.** If the original CAP eliminates weights in greedy optimal order, FastCAP prunes weights in a fixed order, similar to SparseGPT [13]. This reduces computational complexity.

The full pseudocode for FastCAP is provided in Algorithm 2. The same procedure is applied for all weights. We use the following notation:

- $B$ - Fisher block size

- $b$ - block size in iterative pruning algorithm

- $N$ - number of gradients

- $N_B$ - number of blocks per parameter

The Fisher matrix is the same for all blocks, reducing space complexity to $\mathcal{O}(B^2)$. We adapted the trick of maintaining the Fisher inverse in Cholesky form from SparseGPT [13], to account for iterative elimination of the weights in a fixed order. This allows running the iterative process efficiently.

**Algorithm 2** Fast CAP pruning algorithm

1: **1. Fisher accumulation**
2: $F \leftarrow$ matrix with zeros with shape $(B, B)$
3: **for** i = 1:N do **do**
4:    reshape $\nabla_w \mathcal{L}_i$ to $(N_B, B)$
5:    $F = F + \frac{1}{N}\nabla_w \mathcal{L}_i^T \nabla_w \mathcal{L}_i$
6: **end for**
7: **2. Iterative pruning**
8: $M \leftarrow 1_{N_B \times B}$ // binary pruning mask
9: $E \leftarrow 0_{N_B \times b}$ // block pruning errors
10: $F \leftarrow \text{Cholesky}(F^{-1})$ // efficient form of the Fisher inverse
11: **for** $i = 0, b, 2b, \ldots$ **do**
12:    $M_{:,j:(j+b)} \leftarrow$ mask of (1 - p)% weights $w_k$ in $W_{:,j:(j+b)}$ with largest $w_k^2/[F^{-1}]_{kk}^2$
13:    **for** $j = i, \ldots, i + b - 1$ **do**
14:       $E_{:,j-i} \leftarrow W_{:,j}/[F^{-1}]_{jj}$ // pruning error
15:       $E_{:,j-i} \leftarrow (1 - M_{:,j})E_{:,j-i}$ // freeze weights
16:       $W_{:,j:(i+b)} \leftarrow W_{:,j:(i+b)} - E_{:,j-i}F_{j,j:(i+B)^{-1}}$ // update remaining weights to correct output
17:    **end for**
18: **end for**
19: $W \leftarrow W \cdot M$ // set pruned weights to 0

To validate performance of FastCAP we applied it for pruning of large ViT models in a one-shot setting, specifically `eva_giant_patch14_224.clip_ft_in1k` from the TIMM library [48]. This model has >1B parameters. We compared the top-1 accuracy on ImageNet of FastCAP vs Magnitude pruner. One can see that FastCAP significantly outperforms the baseline. At the same time, FastCAP has a reasonable runtime and memory footprint at this scale. The pruning step for the whole model takes 400 seconds on a single A100 GPU.

Table 4: Performance of large ViT after one-shot pruning.

| Model | Sparsity | Method | Accuracy (%) |
|-------|----------|--------|--------------|
| EVA ViT-G | 0 | - | 88.7 |
| | 50 | Magnitude | 87.9 |
| | | FastCAP | **88.1** |
| | 60 | Magnitude | 85.5 |
| | | FastCAP | **86.3** |
| | 70 | Magnitude | 64.3 |
| | | FastCAP | **76.1** |

## C   Training details

**Augmentation/regularization recipe**
For the gradual pruning experiments (with 300 epochs) we have used cyclic learning schedule, with high learning rate directly after the pruning step with gradual decrease up to the next pruning step. For both DeiT-Tiny and DeiT-Small model during the additional fine-tuning for 100 epochs we've applied cosine annealing schedule with $\eta_{\max} = 5 \cdot 10^{-5}, \eta_{\min} = 1 \cdot 10^{-5}$ and all other parameters the same as in the Table 6.

## D   Post-Pruning Recovery

The choice of augmentation parameters and learning rate schedule is critical for high performance. For example, reducing the level of augmentation during fine-tuning for smaller models, e.g. DeiT-Tiny, significant improves performance, whereas larger models, e.g. the 4x larger DeiT-Small, requires strong augmentations for best results even during fine-tuning. See Figure 5 for an illustration; the augmentation procedure is described in detail in C.

Table 5: Summary of the augmentation and regularization procedures used in the work.

| Procedure | DeiT | light1 |
|---|---|---|
| Weight decay | 0.05 | 0.03 |
| Label smoothing $\varepsilon$ | 0.1 | 0.1 |
| Dropout | ✗ | ✗ |
| Stoch.Depth | 0.1 | 0.0 |
| Gradient Clip. | ✗ | 1.0 |
| H.flip | ✓ | ✓ |
| RRC | ✓ | ✓ |
| Rand Augment | 9/0.5 | 2/0.5 |
| Mixup alpha | 0.8 | 0.0 |
| Cutmix alpha | 1.0 | 0.0 |
| Erasing prob. | 0.25 | 0.0 |
| Erasing count | 1 | 0 |
| Test crop ratio | 0.9 | 0.9 |

Table 6: Hyperparameters of the schedules used in gradual pruning.

| Model | Prune freq | LR sched $\{f_{\text{decay}}, \eta_{\max}, \eta_{\min}\}$ | Augm | Batch size | Epochs |
|---|---|---|---|---|---|
| DeiT-Tiny | 20 | $\{\text{cyclic\_linear}, 5 \cdot 10^{-4}, 1 \cdot 10^{-5}\}$ | *light1* | 1024 | 300 |
| DeiT-Small | 20 | $\{\text{cyclic\_linear}, 5 \cdot 10^{-4}, 1 \cdot 10^{-5}\}$ | *deit* | 1024 | 300 |

Moreover, the choice of cyclic learning rate (LR) schedule is critical as well. To illustrate this, we compare convergence obtained when using a *cosine annealing* schedule, which is very popular for pruning CNNs [25, 39, 33], from $\eta_{max} = 5 \cdot 10^{-4}$ to $\eta_{min} = 10^{-5}$, while performing pruning updates 2 times more frequently (one update per 10 epochs) than in our standard setup from the following section 4.2. The results are provided in Figure 5, where cosine annealing (no cycles) is in red. All experiments use the CAP pruner, and highlight the importance of the learning rate and augmentation schedules for recovery.

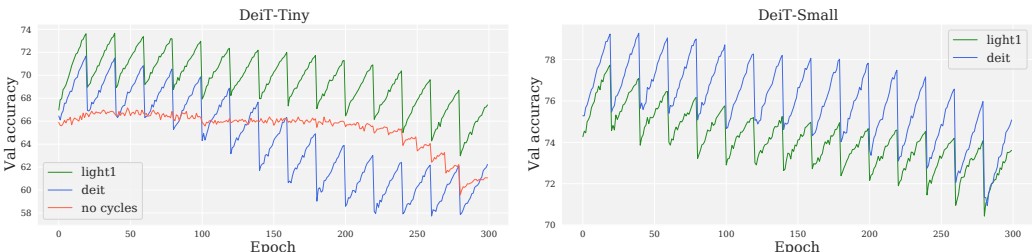

Figure 5: Ablations of the training setting on DeiT-Tiny (up) and DeiT-Small (down). Green curves correspond to the *light1* [41] augmentation recipe, blue curves to the *deit* [43] recipe. The red curve follows training with a single (acyclic) cosine annealing schedule, as in [25, 39].

# E   Additional Results for One-Shot Pruning

In this section we present comparison of Global Magnitude (GM), WoodFisher (WF) and Correlation Aware (CAP) pruners in one-shot pruning setting for DeiT models [43] of different size (i.e DeiT-Tiny, DeiT-Small, DeiT-Base) to study the scaling behavior of ViT sparsification.

Notice, that the gap between magnitude pruning and second order methods is very pronounced for all models, whereas the difference in performance between CAP and WF decreases with increase of the size of model. Nevertheless, CAP performs still noticeably better than WF, especially in high sparsity regime.

**Pruning and finetuning.** In most of the practical setups one cannot achieve both high compression rate and maintain performance of the dense model in one-shot setup. The full retraining procedure allows to achieve high sparsity but is rather expensive in the terms of compute. One can be interested

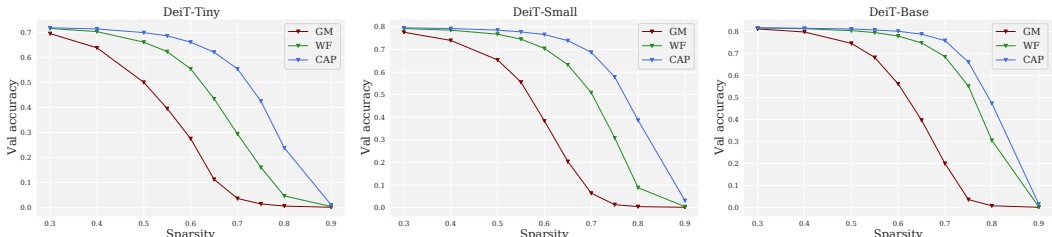

Figure 6: One-shot pruning of different DeiT versions.

to have something in between - moderately sparse model, close in performance to the original model but without the need of long and expensive training.

In the experiments below we prune all models to $50\%$ sparsity, and fine-tune for 20 epochs. In addition to ViT/DeiT, we also consider similar models based on variants of self-attention [28, 1], and compare against a GM baseline. We use a linearly-decaying learning rate schedule between $\eta_{max} = 10^{-4}$ to $\eta_{max} = 10^{-5}$ and the DeiT training recipe [43]. The results are given in Table 7, and show that CAP can almost fully-recover accuracy in this setup for all models; the gaps from GM and WF (see DeiT-Small 75 and 90%) are still very significant.

Table 7: One-shot + fine-tuning on ImageNet-1k.

| Model | Method | Sparsity (%) | Top1-Accuracy (%) |
|---|---|---|---|
| | Dense | 0 | 79.8 |
| | GM | 50 | 79.0 |
| | CAP | | **79.5** |
| DeiT-Small | GM | 75 | 74.3 |
| | WF | | 75.8 |
| | CAP | | **76.9** |
| | GM | 90 | 45.6 |
| | WF | | 59.3 |
| | CAP | | **65.1** |
| | Dense | 0 | 81.8 |
| | GM | 50 | 81.5 |
| | CAP | | **81.6** |
| DeiT-Base | GM | 75 | 80.1 |
| | WF | | 80.2 |
| | CAP | | **81.0** |
| | GM | 90 | 68.1 |
| | WF | | 69.2 |
| | CAP | | **76.3** |

| Model | Method | Sparsity (%) | Top1-Accuracy (%) |
|---|---|---|---|
| | Dense | 0 | 83.1 |
| | GM | 50 | 82.5 |
| | WF | | 82.5 |
| | CAP | | **82.8** |
| ConvNext-Small | GM | 75 | 80.7 |
| | WF | | 81.0 |
| | CAP | | **81.9** |
| | GM | 90 | 70.9 |
| | WF | | 73.2 |
| | CAP | | **78.2** |
| | Dense | 0 | 82.0 |
| XCiT-Small | GM | 50 | 81.7 |
| | CAP | | **81.9** |
| | Dense | 0 | 81.3 |
| Swin-Tiny | GM | 50 | 80.6 |
| | CAP | | **80.9** |

# F  Experiments with other models

In the main part of the text, we considered only gradual pruning of ViT models, but the proposed method is applicable to any architecture for image classification, such as convolutional neural network (CNN) or a ViT-CNN hybrid. We have selected recently proposed EfficientFormer [27] as a member of ViT-CNN hybrid family and trained it using the same setting and hyperparameters as for DeiT-Small. Two CNN architectures - ResNet50-D [2] and EfficientNetV2-Tiny [42] [3], considered in this work were trained with the use of augmentation and regularization procedure described in the recent PyTorch blog post. Differently from most of the prior art we have used the ResNet50-D trained with the modern recipe from timm repository.

For ResNet50-D we prune all convolutional weights except the first convolution and we keep the classification layer dense. In EfficientNetv2-Tiny we do not prune depthwise convolutions since they do not contribute much to the total number of parameters and FLOPs but they are important to the model performance. We have set the block size to be 256 for ResNet50-D and 16 for EfficientNetV2-Tiny while keeping all the other hyperparameters of CAP the same as for DeiT experiments. Such a small block size was chosen for EfficientNetV2-Tiny due to the fact that it is the largest common divisor of the prunable weights.

First of all, we conducted comparison between one-shot pruning methods for ResNet50-D. We compare between Uniform and Global magnitude pruning, WoodFisher with block size of 256,

---

[2]`resnet50d` checkpoint with 80.5 % accuracy for dense model
[3]`efficientnetv2_rw_t` checkpoint with 82.3 % accuracy for dense model

M-FAC with block size of 2048 and CAP with uniform and global sparsity. One can observe that CAP outperforms all previous methods even when comparing uniform sparsity with global sparsity. Contrary to the case of DeiT where there is no much difference in performance between uniform and global magnitude pruning for ResNet50-D global sparsity turns out to be much better. This results is quite expectable since CNN are not uniform and deeper layers are mode wide than those close to the input.

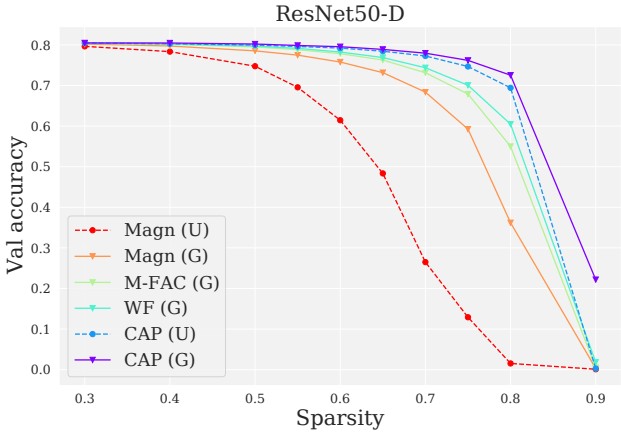

Figure 7: One-shot pruning of ResNet50-D model on ImageNet dataset.

Next we carried out one-shot + finetuning experiments with ResNet50-D keeping setup the same as for DeiT models in Table 7. We have selected 75 % and 90 % as the difference between methods becomes pronounced only at sufficiently high sparsity. Notably, the performance of Global magnitude and WF is roughly the same, the initial difference after one-shot pruning between WF and Global Magnitude vanishes during the finetuning procedure, whereas there is still a gap in performance between CAP and other methods.

Table 8: One-shot + fine-tuning on ImageNet.

| Model | Method | Sparsity (%) | Top1-Accuracy (%) |
|---|---|---|---|
| | Dense | 0 | 80.5 |
| | GM | | 79.0 |
| | WF | 75 | 79.0 |
| ResNet-50D | CAP | | **79.2** |
| | GM | | 74.7 |
| | WF | 90 | 74.8 |
| | CAP | | **75.2** |

Finally we conducted gradual pruning runs following the same sparsification schedule as for DeiT-models. The EfficientFormer and EfficientNet models despite being already very optimized and parameter efficient can be still compressed with small drop in accuracy.

# G   Scaling behaviour of pruning with respect to model size

To study the scaling behavior with respect to model size we took all variants of the ConvNext2 family of models [49] since it covers wide range of model sizes except for the large one due to the memory and compute constraints. The smallest model from the family - ConvNext2-Atto has 3.7M parameters whereas the largest considered ConvNext2-Large has 198M parameters. All the models were pruned to 50% in one-shot. We observed that the relative accuracy drop (difference between accuracy of the dense and sparse model) initially decreases with increase of model size and then reaches a plateau. CAP consisently outpeforms WF across all scales and the difference is the most pronounced for the smallest model.

Table 9: Gradual pruning on ImageNet. Parentheses followed by the upwards directed arrow denote additional fine-tuning for 100 epochs.

| Model | Method | Sparsity (%) | Top1-Accuracy (%) |
|---|---|---|---|
| EffFormer-L1 | Dense | 0 | 78.9 |
| | CAP | 50 | 78.0 |
| | | 60 | 77.4 |
| | | 75 | 76.4 |
| | | 90 | 72.4 (72.8 ↑) |
| ResNet-50D | Dense | 0 | 80.5 |
| | CAP | 50 | 79.8 |
| | | 60 | 79.7 |
| | | 75 | 79.2 (79.6 ↑) |
| | | 90 | 77.1 (77.5 ↑) |
| EffNetV2-Tiny | Dense | 0 | 82.4 |
| | CAP | 50 | 81.0 |
| | | 60 | 80.6 |
| | | 75 | 79.6 (80.0 ↑) |
| | | 90 | 75.0 |

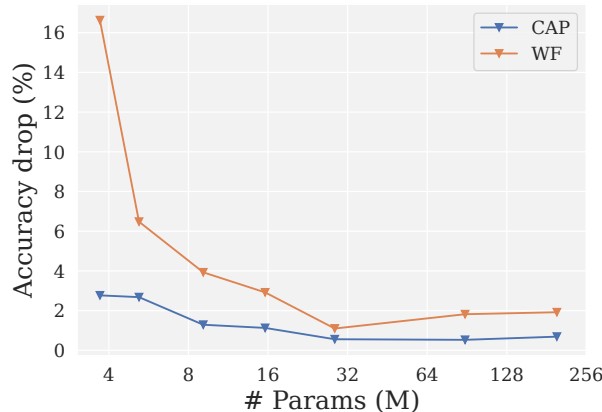

Figure 8: **Left**: CAP vs WoodFisher for pruning of ConvNext2 family.

## H   Timings

Any algorithms involving second order loss information are believed to require tremendous amounts of compute. Time required for calculation of pruning scores and the OBS update comprises collection of grads, Fisher inverse rank-1 updates and additional pruning iteration for CAP. We have measured the time of single pruning step for DeiT-Small and present the results in Table 10. All measurements were done on a single RTX A6000 GPU with 48GB of memory. One can observe that the amount of time needed to perform a pruning update is not very large, especially when compared to the duration of typical training procedure of modern computer vision models on ImageNet that usually takes several days on a multi-gpu node. Note that the additional step for CAP adds small fraction of total computational time relative to other steps of the OBS method.

Table 10: Minutes per pruning step for DeiT-Small.

| Model | Method | Time (minutes) |
|---|---|---|
| DeiT-Small | Fast WoodFisher [23] | 20 |
| | CAP | 23 |

# I Composite compression

In addition to weight pruning one can decrease storage and inference cost with the help of other compression approaches: quantization (casting weights and activations to lower precision) and token pruning specific for the transformer architecture.

## I.1 Quantization-Aware Training

Weight quantization is done in the following way - one takes sparse checkpoint and then runs quantization aware training (QAT). We ran QAT training for 50 epochs with linearly decaying learning rate schedule from $\eta_{\max} = 10^{-4}$ to $\eta_{\min} = 10^{-5}$. Models are quantized to 8-bit precision. In all experiments performed accuracy of quantized model almost reproduces the accuracy of the sparse model stored in full precision.

Table 11: ImageNet-1K top-1 accuracy for sparse models after QAT training.

| Model | Sparsity (%) | Accuracy (%) |
|---|---|---|
| DeiT-Tiny | 75 | 72.2 |
| DeiT-Small | 75 | 77.7 |
| DeiT-Base | 75 | 81 |

## I.2 Token Pruning

There are different approaches for token pruning proposed in the literature. In this work we follow the one from [35]. Specifically, in DynamicViT one selects the ratio of tokens being pruned at each step with the lowest importance score, predicted by the model itself. Following the main setup from the paper we prune tokens after $3^{\mathrm{rd}}$, $6^{\mathrm{th}}$, $9^{\mathrm{th}}$ block, and the token pruning ratio after each block is $\rho = 0.2$ (i.e 20% least improtant tokens are pruned).

Table 12: ImageNet-1K top-1 accuracy for sparse models with token pruning.

| Model | Method | Sparsity (%) | Top1-Accuracy (%) |
|---|---|---|---|
| DynamicViT-Tiny | CAP | 50 | 72.0 |
| | | 60 | 71.6 |
| | | 75 | 70.2 |
| DynamicViT-Small | CAP | 50 | 79.5 |
| | | 60 | 79.4 |
| | | 75 | 78.7 |

## I.3 Semi-structured sparsity

While CPUs can utilize sparsity patterns of arbitrary form to speed-up the computations at the present time modern GPU accelerators can handle only restricted form of unstructured sparsity, namely the $N : M$ sparsity pattern that enforces exactly $N$ non-zero values for each block of $M$ weights. Namely, since the introduction of Ampere architecture NVIDIA GPUs have special kernels that can work with $2 : 4$ sparse matrices [30]. One can integrate the $N : M$ sparsity in the CAP framework without significant changes. The only difference with the original CAP approach is that while running the CAP iterations one doesn't prune a given weight in case in a group of $M$ weights to which this weights belongs to there are $M - N$ zero weights. Since the sparsity pattern is significantly constrained compared to generic unstructured sparsity pattern drop in performance after doing pruning step and consequent fine-tuning is more challenging than it would be for unconstrained sparsity. In experiments below we prune models to $2 : 4$ sparsity either in one-shot setting and one-shot+finetune. We apply shorter (10 epochs) and longer (50 epochs) finetuning procedure with linearly decaying learning rate schedule. According to the results in the Table 13 CAP significantly outperforms competitive methods for one-shot pruning, although the drop in performance is quite large for all methods. After finetuning procedure difference between different methods decreases. Nevertheless, there is some gap in performance between second order methods and magnitude pruning even after relatively long finetuning.

To demonstrate practical benefits from 2:4 sparsity pattern we compiled both sparse and dense models via TensorRT engine and compared the throughput. The inference was executed on Nvidia T4 GPU

Table 13: Semi-structured $2:4$ pruning of ViT models.

| Model | Method | Epochs | Top1-Accuracy (%) |
|---|---|---|---|
| | Dense | | 72.2 |
| | GM | 0 | 24.4 |
| | WF | | 44.1 |
| | CAP | | **55.9** |
| DeiT-Tiny | GM | 10 | 68.8 |
| | WF | | 71.1 |
| | CAP | | **71.5** |
| | GM | 50 | 72.5 |
| | WF | | **72.7** |
| | CAP | | **72.7** |
| | Dense | | 79.8 |
| | GM | 0 | 53.6 |
| | WF | | 67.8 |
| DeiT-Small | CAP | | **72.0** |
| | GM | 10 | 77.9 |
| | WF | | **78.1** |
| | CAP | | 78.0 |
| | GM | 50 | 78.6 |
| | WF | | **79.0** |
| | CAP | | **79.0** |
| | Dense | | 81.8 |
| | GM | 0 | 66.4 |
| | WF | | 73.7 |
| | CAP | | **78.1** |
| DeiT-Base | GM | 10 | 81.2 |
| | WF | | **81.3** |
| | CAP | | **81.3** |
| | GM | 50 | **81.7** |
| | WF | | 81.6 |
| | CAP | | **81.7** |

with batch size of 64 in half precision. Sparsity allows for small but certain speedup for models of different scale.

Table 14: Speedup factors for $2:4$ sparsity.

| Model | Speedup |
|---|---|
| DeiT-Tiny | 1.07 |
| DeiT-Small | 1.07 |
| DeiT-Base | 1.10 |

## J  CAP/WF hyperparameters

Following the oBERT's directions [23] on identifying the optimal set of hyperparameters via one-shot pruning experiments, we conduct a grid search over the three most important hyperparameters:

- Number of grads collected for Fisher inverse
- Dampening constant $\lambda$
- Block size

The more grads are collected, the more accurate is the empirical Fisher inverse estimate, however, more compute is required at the same time. We chose $N = 4096$ as a point from which further increase of Fisher samples doesn't improve performance a lot. Dependence of the one-shot pruning performance at different sparsities vs number of grads is presented on Figure 9.

The next parameter to be studied is the dampening constant $\lambda$ in. This constant regularizes the empirical Fisher matrix and allows to avoid instabilities in computation of the inverse. However, this constant decreases the correlation between different weights and in the limit $\lambda \to \infty$ OBS reduces to magnitude pruning. The optimal dampening constant for CAP ($\lambda_{\mathrm{opt}} = 10^{-8}$) is smaller than the one for WoodFisher ($\lambda_{\mathrm{opt}} = 10^{-6}$), i.e CAP remains numerically and computationally stable with smaller amount of regularization compared to WF (we observed that for $\lambda < 10^{-7}$ WF performance starts to deteriorate rapidly).

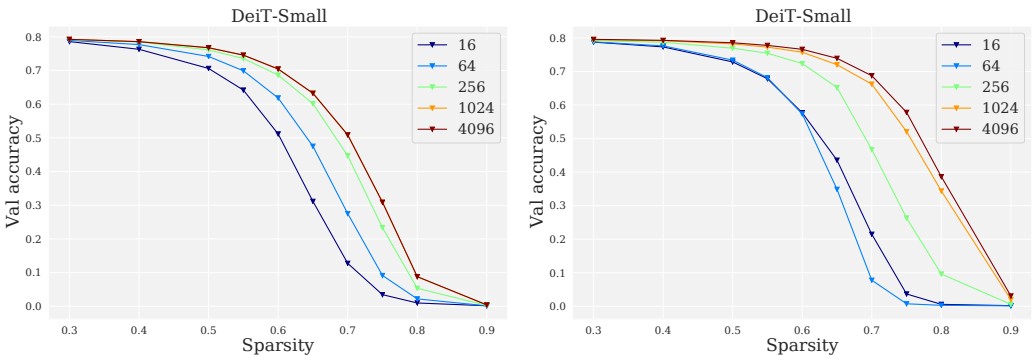

Figure 9: **Left**: One-shot pruning performance of WoodFisher. **Right**: One-shot pruning performance of CAP.

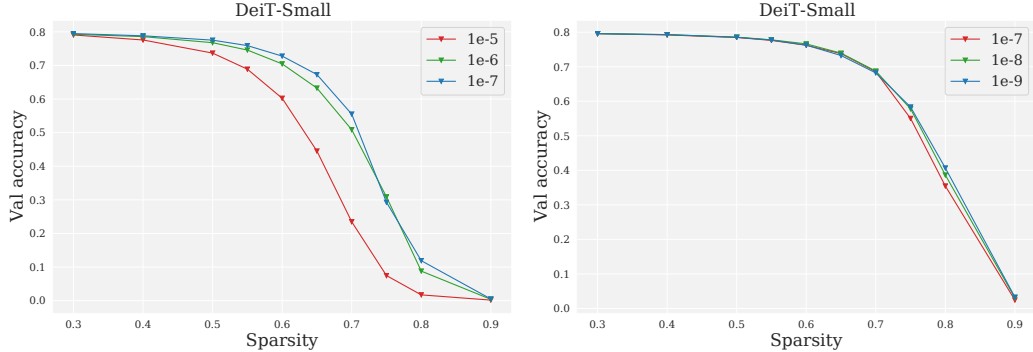

Figure 10: **Left**: One-shot pruning performance of WoodFisher. **Right**: One-shot pruning performance of CAP.

And the last but not the least important parameter is the block size in [39]. The larger the block size is, the more correlations between different weights are taken into account. However, as mentioned in 2 the computational and storage cost scales with the block size. Moreover, for a fixed number of gradients in the Fisher estimate matrix with larger block sizes is likely to be worse conditioned. Therefore, one would like to work with smaller block sizes but not to keep the approximation as accurate as possible. We've selected block size according to the accuracy-efficiency trade-off.

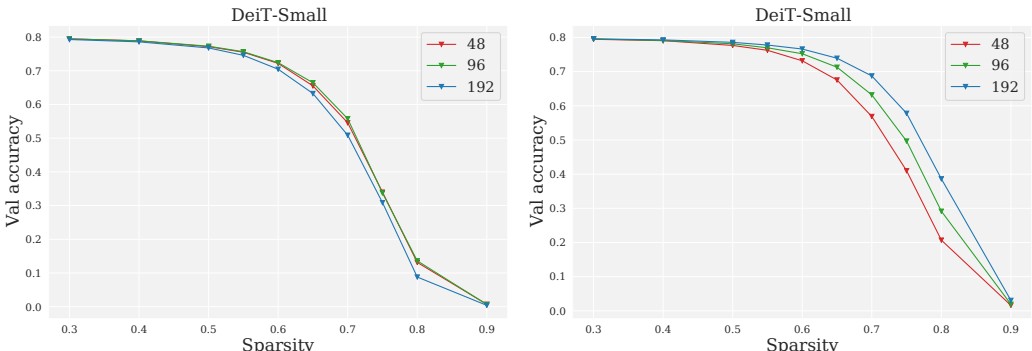

Figure 11: **Left**: One-shot pruning performance of WoodFisher. **Right**: One-shot pruning performance of CAP.

In addition, we've studied the benefit from application of multiple recomputations in the one-shot pruning setting for WoodFisher and CAP. Since the assumption of static Fisher matrix $\mathbf{F}(\mathbf{w}^*)$ doesn't hold in general, we expect that multiple recomputations are likely to result in higher one-shot accuracy

in accordance with the result from [14]. This is indeed the case. The gain from recomputations is more pronounced for WoodFisher, since CAP already performs implicit Fisher inverse updates in its operation. Yet, the efect is not vanishing even for the case of CAP.

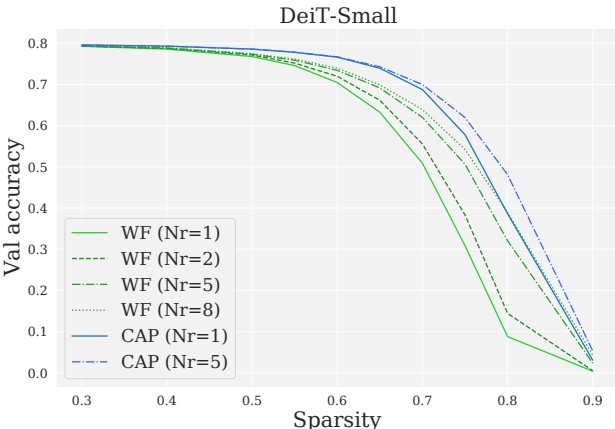

Figure 12: One-shot performance for WF and CAP with different number of recomputations $N_r$.

## K    Details and hyperparameter choices for other pruning methods

In this section we provide some additional details about methods compared on Figures 2 and 7. The popular Movement Pruning [38] computes the weight saliency scores during the training procedure, hence it is not a one-shot pruning method by the definition. We have observed that use of the naive elementwise product of gradient and weights ( i.e $\rho_i = w_i \odot \nabla_{w_i} \mathcal{L}(w)$) leads to a poor performance, significantly below even the Magnitude Pruning baseline. However, the following first order pruning criterion:

$$\rho_i = \sum_{k=1}^{N} \|w_i^{(k)} \odot \nabla_{w_i} \mathcal{L}^{(k)}(w)\| \tag{9}$$

allows to get reasonable saliency scores that produce more accurate sparse models than Magnitude Pruning. However, its performance is still inferior to any of the second order pruning methods. This method is denoted as GrW (Gradient times weight) on Figures 2 and 7.

M-FAC Pruner proposed in [14] is a pruner leveraging second order information that doesn't require an explicit construction of Fisher Inverse matrix. Therefore, unlike WoodFisher and CAP that require $O(Bd)$ memory computation and storage cost of this method is constant with respect to the block size and one can take into account correlations between larger groups of weights for free. Following the original paper we chose block size of 2k as the best performing one. However, one can see from Figures 2 and 7 that smaller block sizes turn out to perform better. A possible explanation of this phenomenon is that the Fisher Inverse estimate becomes too noisy and unstable for large blocks.

## L    Execution latency

In addition to the plot throughput vs accuracy shown in the main part we present in this section execution latency per sample vs latency when running models on the DeepSparse engine. The results are presented on Figure 13.

## M    Comparison with AC/DC training

In addition to the sparse training from scratch with periodic updates of the sparsity weights with some saliency criterion for weight elimination and regrowth [10] one can consider alternating compressed/decompressed training (AC/DC), proposed in [33]. Namely one switches between dense stages with standard unconstrained training of the model, and sparse stages when the model is pruned to the target sparsity level and trained with the frozen sparsity mask until the beginning of the next dense stage, when the sparsity mask is removed. This procedure produces both accurate dense and sparse models.

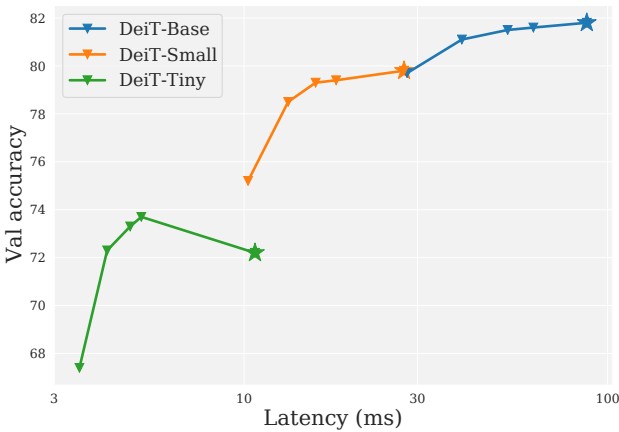

Figure 13: Accuracy vs latency on ImageNet-1k.

Following the original paper we use magnitude pruning as a saliency criterion for weight pruning. The augmentation and regularization pipeline follows the settings from [43]. All models with AC/DC were trained for 600 epochs in total with first pruning step at epoch 150 followed by 7 sparse stages 25 epochs long each, and 6 dense stages of the same length. The last dense stage lasts 50 epochs and the last sparse is 75 epochs long. Learning rate is gradually decayed from $\eta_{max} = 5 \cdot 10^{-4}$ to $\eta_{min} = 10^{-6}$ with cosine annealing. Initial warm-up phase with linearly increasing learning rate is 20 epochs. We compare AC/DC with CAP models finetuned for additional 100 epochs.

Table 15: AC/DC vs CAP (finetuned for additional 100 epochs) on ImageNet-1k.

| Model | Method | Sparsity (%) | Top1-Accuracy (%) |
|-------|--------|--------------|-------------------|
| DeiT-Small | CAP | 60 | 79.9 |
| | AC/DC | | **80.4** |
| | CAP | 75 | **79.0** |
| | AC/DC | | **79.0** |
| | CAP | 90 | **75.8** |
| | AC/DC | | 72.0 |

One can observe that at low sparsity AC/DC achieves higher accuracy for the same sparsity target (even outperforming the dense baseline by 0.6%), whereas for 75% performance of both methods is equal, and CAP outperforms AC/DC at higher sparsity. However, one should note, that CAP uses computational budget (including the training of original model) of 440 epochs for 60% sparsity, 520% for 75% and 700% for 90% vs 600 epochs used in AC/DC.

# N    One-shot pruning of DETR

The approach presented in the paper is not limited to the image classification task, but can be applied to other computer vision tasks, such as object detection. We chose the DeTR model [4] with ResNet50 backbone and ran one-shot pruning procedure with global magnitude, WoodFisher and CAP pruner. Specifically, we pruned all convolutional layers in the CNN backbone expect the first one and all linear projections in transformer encoder and decoder blocks while keeping the detection heads dense. The results are presented in Table 16. Following the standard protocol we used bbox mAP for evaluation. One can observe, that difference between the second order methods and magnitude pruning is very pronounced even for relatively small sparsity of 50%, and CAP outperforms WF pruner.

Table 16: One-shot pruning of DeTR.

| Model | Method | Sparsity (%) | bbox mAP |
|-------|--------|--------------|----------|
|       | Dense  | 0            | 0.42     |
| DeTR  | GM     |              | 0.16     |
|       | WF     | 50           | 0.36     |
|       | CAP    |              | **0.38** |

## O  Proof of Theorem 1

**Theorem O.0.** *Let $\mathcal{S}$ be a set of samples, and let $\nabla_{\ell_1}(\mathbf{w}^*), \ldots, \nabla_{\ell_m}(\mathbf{w}^*)$ be a set of gradients with $i \in \mathcal{S}$, with corresponding empirical Fisher matrix $\widehat{\mathbf{F}}^{-1}(\mathbf{w}^*)$. Assume a sparsification target of $k$ weights from $\mathbf{w}^*$. Then, a sparse minimizer for the the constrained squared error problem*

$$min_{\mathbf{w}'} \frac{1}{2m} \sum_{i=1}^{m} \left( \nabla_{\ell_i}(\mathbf{w}^*)^\top \mathbf{w}' - \nabla_{\ell_i}(\mathbf{w}^*)^\top \mathbf{w}^* \right)^2 \text{ s.t. } \mathbf{w}' \text{ has at least } k \text{ zeros}, \quad (10)$$

*is also a solution to the problem of minimizing the Fisher-based group-OBS metric*

$$argmin_{Q,|Q|=k} \frac{1}{2} \cdot \mathbf{w}_{\mathbf{Q}}^{*\top} \left( \widehat{\mathbf{F}}^{-1}(\mathbf{w}^*)_{[Q,Q]} \right)^{-1} \mathbf{w}_{\mathbf{Q}}^*. \quad (11)$$

*Proof.* We start by examining the unconstrained squared error function in Equation (10), which we denote by $\mathcal{G}$. Clearly, the function $\mathcal{G}$ is a $d$-dimensional quadratic in the variable $\mathbf{w}'$, and has a minimum at $\mathbf{w}^*$. Next, let us examine $\mathcal{G}$'s second-order Taylor approximation around $\mathbf{w}^*$, given by

$$(\mathbf{w}' - \mathbf{w}^*)^\top \left( \frac{1}{m} \sum_{i=1}^{m} \nabla_{\ell_i}(\mathbf{w}^*)^\top \nabla_{\ell_i}(\mathbf{w}^*) \right) (\mathbf{w}' - \mathbf{w}^*), \quad (12)$$

where we used the fact that $\mathbf{w}^*$ is a minimum of the squared error, and thus the function has 0 gradient at it. However, by the definition of the empirical Fisher, this is exactly equal to

$$(\mathbf{w}' - \mathbf{w}^*)^\top \widehat{\mathbf{F}}(\mathbf{w}^*)(\mathbf{w}' - \mathbf{w}^*). \quad (13)$$

The Taylor approximation is exact, as the original function is a quadratic, and so the two functions are equivalent. Hence, we have obtained the fact that, under the empirical Fisher approximation, a $k$-sparse solution minimizing Equation 10 will also be a $k$-sparse solution minimizing Equation 1. However, the question of finding a $k$-sparse solution minimizing Equation 1 is precisely the starting point of the standard OBS derivations (see e.g. [39] or [23]), which eventually lead to the formula in Equation (11). This concludes the proof. □

## P  Augmentation choice for Empirical Fisher

We compared the performance of CAP with Empirical Fisher computed on image-label pairs where the validation transforms were applied to images (i.e center crop with resize) and the same set of augmentations used for training and finetuning (RandAugment transforms, Label smoothing, e.t.c.). We observed that the sparsity solution obtained without augmenting samples for Empirical Fisher estimate turns out to be strongly overfitting. We point out that in both cases we use the same population size for Empirical Fisher.

Figure 14 illustrates this result: using validation augmentation (red) yields better training loss but degenerates in terms of validation accuracy. A possible explanation is that CAP chooses an overfitting solution which the model is unable to escape during finetuning.

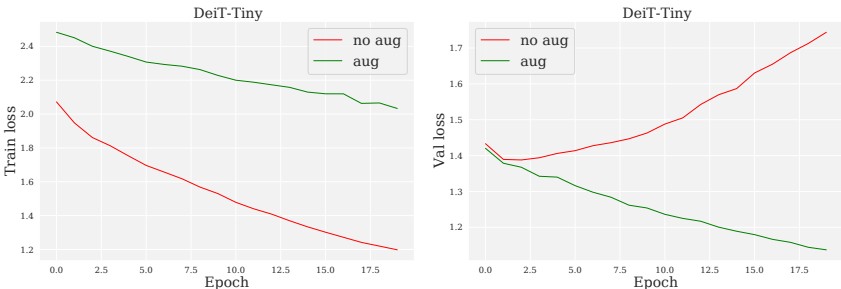

Figure 14: Training (**left**) and validation loss (**right**) for one-shot + finetuning.

## Q Fisher matrix structure

In order to validate the necessity of taking into account the correlations between weights, one has to make sure that the empirical Fisher matrix used as proxy for Hessian is non-diagonal. We have visualized an average block of empirical Fisher for a particular layer on Figure 15 from DeiT-Tiny and ConvNext-Small models. One can see, Fisher matrix exhibits a pronounced non-diagonal structure, which justifies the need of a careful and thorough treatment of weight correlations.

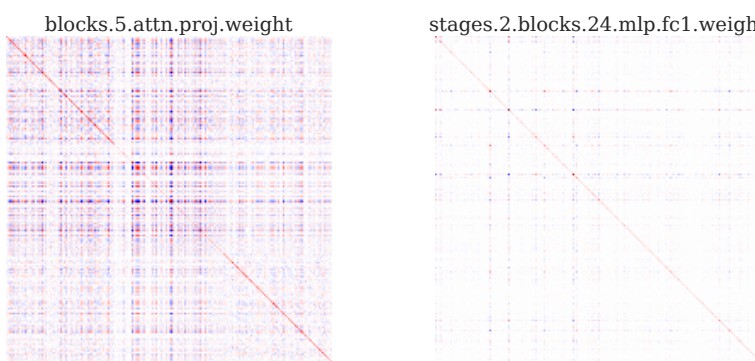

Figure 15: **Left**: empirical Fisher block for a weight from DeiT-Tiny. **Right**: empirical Fisher block for a weight from ConvNext-Small.

## R Broader impact

Compressed models are not expected to exhibit more malicious and potentially harmful behavior compared to the dense models. However, they may face the same issues like the original models in safety-critical applications such as susceptibility to adversarial attacks and distribution shifts.

## S Limitations

The proposed method is mostly suitable for small and medium sized models (up to order of $\sim 100M$). For larger models the compute and storage cost associated with the estimate of empirical Fisher becomes prohibitively expensive. Compression of the largest models considered is this paper requires 2-4 high-end GPUs (A100 with 80GiB). Pruning models to high sparsity requires significant amount of training. Search for fast and efficient procedures for the recovery of compressed models is left as potential direction for further research.

