# OpenReview forum: "CAP:  Correlation-Aware Pruning for Highly-Accurate Sparse Vision Models"
_NeurIPS.cc/2023/Conference — NeurIPS 2023 poster_

### Official Review · Reviewer_gPY6 · 2023-07-02

**Soundness:** 3 good
**Presentation:** 3 good
**Contribution:** 3 good
**Rating:** 5
**Confidence:** 4

**Summary:**

This paper studies the unstructured pruning problem for vision transformer models. The Correlation Aware Pruner (CAP) is proposed. CAP takes into account weight correlations and achieves a new state-of-the-art result.

**Strengths:**

- The proposed method achieves strong experimental results. For the first time, ViT models can attain high sparsity levels (75-80%) without significant accuracy loss (<1%). Note that previous methods reached at most 50% sparsity.
- CAP is reasonable and well-motivated. This paper points out an important problem that removed weights may themselves be correlated. It also highlights the importance of the learning rate schedule. Overall, this paper provides lots of useful information for unstructured pruning for ViT.
- The whole paper is organized and written well. Experiments are sufficient and sound.

**Weaknesses:**

- The biggest problem is unstructured pruning struggles with GPU devices. The authors test models’ latency on a sparsity-aware CPU inference engine. However, GPU is not considered. Overall, unstructured pruning performs badly on GPU which limits its practical ability.
- Some results of SViTE are missing in Tab. 2. It is better to change the order to put CAP at the bottom of each section.

**Questions:**

Please see weaknesses.

**Limitations:**

The authors have adequately addressed the limitations

---

> ### Author Rebuttal · Authors · 2023-08-09
>
> We thank the reviewer for the feedback!
>
> **1. Applicability to GPUs:**
>
> In Appendix H.3, we provided results for pruning with hardware-supported 2:4 sparsity pattern [1]. This semi-structured sparsity pattern can lead to speedups of tensor operations on the Ampere and Hopper GPU architecture. One can see that our method allows us to compress the models without significant drop in performance in this setup. For convenience, we present a sample of the results in **Table 1** below.
>
> **Table 1. Semi-structured 2:4 pruning of ViT models followed by 10 epochs of finetuning.**
>
> |    Model    | Method | Top1-Accuracy (\%) |
> |:-----------:|:------:|:------------------:|
> |  DeiT-Tiny  |    -   |        72.2        |
> |             |   GM   |        68.8        |
> |             |   CAP  |        71.5        |
> |  DeiT-Small |    -   |         79.8       |
> |             |    GM  |         77.9       |
> |             |   CAP  |         79.0       |
> |   DeiT-Base |    -   |         81.8       |
> |             |    GM  |         81.2       |
> |             |   CAP  |         81.3       |
>
> **2. Comparison with SViTE:**
>
> We included all the results from the original paper (where there was a result for a specific sparsity level) as well as additional experiments we performed ourselves (in case there was no experiment with the given sparsity in the paper). The training procedure in SViTE requires 2x more epochs than the full CAP pipeline, and has to be rerun for every sparsity level. (Recall that CAP produces all sparse models in a single run.) Due to the high computational cost of retraining ViT models, we were unable to perform experiments for some sparsity levels.
>
> We would have liked to run these additional experiments for SViTE for the rebuttal phase, but unfortunately the experiments wouldn’t finish by the end of the rebuttal period, given the large cost of running 600 epochs of training for large models on our computational resources. We plan to add a full comparison, across all sparsity levels, in the next revision. At the same time, we note that currently CAP appears to outperform SViTE across all sparsity levels. (Please see Figure 4 for an illustration across the entire DeiT family of models.)
>
> [1] https://developer.nvidia.com/blog/exploiting-ampere-structured-sparsity-with-cusparselt/

---

> > ### Author Response · Authors · 2023-08-21
> >
> > We wish to make an addition to our earlier response:
> >
> > As requested, we ran S-ViTE with 80% sparsity for DeiT-Small. Consistent with the previous results, the proposed CAP methods significantly outperforms S-ViTE.
> >
> > | Sparsity | Method | Top-1 Accuracy (%) |
> > |:--------:|:------:|:------------------:|
> > |    80    | S-ViTE |        75.8        |
> > |          |   CAP  |        78.0        |
> >
> > In addition, we note that S-ViTE is quite computationally expensive. Running of the algorithm for 600 epochs with the same hyperparameters as in the original paper requires 5 days on 4 RTX 3090 GPUs, whereas our approach produces a more accurate model using only half of S-ViTE computational cost.

---

### Official Review · Reviewer_kMpj · 2023-07-05

**Soundness:** 3 good
**Presentation:** 3 good
**Contribution:** 2 fair
**Rating:** 3
**Confidence:** 4

**Summary:**

The submission proposes a second-order method for unstructured pruning of neural net parameters, to leverage the efficiency gains of sparsity. It uses an optimization based on the empirical Fischer matrix to find saliency scores, that are used for the order in which weights are pruned. They present an algorithm based on solving the problem on individual sub-blocks, and then using that within a global pruning. They also present a schedule, across different hyperparameters, for training with pruning. This is demonstrated on ConvNeXt and DeiT models, along with other CNN and ViT models in the supplement.

**Strengths:**

**i)** Achieves better accuracy & sparsity.

The proposed pruning gets pareto-dominant results relative to some similar prior work. The evaluation is done across multiple choices of the sparsity vs accuracy tradeoff.

Not all experiments consider some of the most similar previous methods, though, such WoodFisher in the "Gradual Pruning" results, though the original work did include gradual pruning during training. (See section 5.2 of [38])

**ii)** Considers good selection of recent models

Experiments demonstrate the pruning on both up-to-date ConvNet and vision transformer models. In the supplement and code, experiments on ResNet-50 and EfficientNet are also provided.

**iii)** Includes code

The submission includes the code used to run the experiments, based on SparseML, along with YAML files specifying the hyperparameters for a number of the experiments.

**Weaknesses:**

**iv)** Largely superseded by Optimal BERT Surgeon in a lot of settings

OBS is a more scalable algorithm based on the same underlying principles. The core contribution seems to be a different method of approximating the solution to the original 2nd-order problem, plus better hyperparameter tuning. It does seem like a reasonable approach for relatively smaller models, to seek a tighter approximation to the original optimization.

**v)** Source of improvement in end metrics remains unclear

At its core, the starting point of the construction of the method in the submission is similar to [38]: namely, second-order pruning with the empirical Fisher matrix in place of the Hessian. The differences from [38] then include a) a different method of approximating the pruning on the full matrix, described toward the end of Section 3.1, and b) improved hyperparameters during gradual pruning.

The paper states that ablation studies are given in appendices B, I, and J. These appendices specify different sets of hyperparameters, and include results of sparsity vs accuracy for multiple different choices of some of these hyperparameters. I didn't find any "ablation" studies that evaluate parts of the proposed method with other parts "ablated" or removed: for example, the results of using Algorithm 1 *without* the changes to data augmentation, the sparsity and learning rate schedule, etc.

Experiments also don't compare against [38] for the gradual pruning case, so the effect of the hyperparameter tuning and schedules isn't measured by a comparison to baseline. It's possible that the improved choices in which weights to prune, as seen in the one-shot case, could also affecting the gradual pruning case, but it's also possible that the training can close a lot of this gap.

The experiments also use very recent ViT-based models, showing some more up-to-date results than prior work. It is plausible that better pruning is necessary to get good results for these models, as argued, but this is incompletely supported.

**Questions:**

**Code**: Some miscellaneous minor comment from trying the included code

README.md marks the step to install wandb as "optional." Not entirely optional, though, given that it does import it:

```
  File "CAP-code/research/one_shot_pruning.py", line 4, in <module>
    import wandb
ModuleNotFoundError: No module named 'wandb'
```

Commenting out the import, then see:

```
AttributeError: module 'numpy' has no attribute 'object'.
`np.object` was a deprecated alias for the builtin `object`. To avoid this error in existing code, use `object` by itself. Doing this will not modify any behavior and is safe.
The aliases was originally deprecated in NumPy 1.20; for more details and guidance see the original release note at:
    https://numpy.org/devdocs/release/1.20.0-notes.html#deprecations
```

within the import of onnx inside sparseml:

```
    from sparseml.pytorch.utils import LOGGING_LEVELS, BaseLogger, LoggerManager
  File "${HOME}/anaconda3/envs/CAP/lib/python3.9/site-packages/sparseml_nightly-0.13.0.20230704-py3.9.egg/sparseml/pytorch/utils/__init__.py", line 23, in <module>
    from .exporter import *
  File "${HOME}/anaconda3/envs/CAP/lib/python3.9/site-packages/sparseml_nightly-0.13.0.20230704-py3.9.egg/sparseml/pytorch/utils/exporter.py", line 26, in <module>
    import onnx
  File "${HOME}/anaconda3/envs/CAP/lib/python3.9/site-packages/onnx-1.10.1-py3.9-linux-x86_64.egg/onnx/__init__.py", line 20, in <module>
    import onnx.helper  # noqa
  File "${HOME}/anaconda3/envs/CAP/lib/python3.9/site-packages/onnx-1.10.1-py3.9-linux-x86_64.egg/onnx/helper.py", line 17, in <module>
    from onnx import mapping
  File "${HOME}/anaconda3/envs/CAP/lib/python3.9/site-packages/onnx-1.10.1-py3.9-linux-x86_64.egg/onnx/mapping.py", line 27, in <module>
    int(TensorProto.STRING): np.dtype(np.object)
  File "${HOME}/anaconda3/envs/CAP/lib/python3.9/site-packages/numpy/__init__.py", line 305, in __getattr__
    raise AttributeError(__former_attrs__[attr])
```

Following instructions in the README had given me `numpy==1.24.3` and `onnx==1.10.1`. It's likely that one needs to hold back numpy, likely an underlying issue with the requirements specified for SparseML. Presumably this codebase is using an old version of sparseml that specifies a requires.txt such that it's versions specified with '>=' no longer really work with the latest versions now available. Authors should check & perhaps update the install commands in their README accordingly, or update their sparseml version. I was able to resolve this with instead:

```
conda install numpy==1.20.3
```

Generally recommended practice for distributing this kind of research code is to explictly specify versions of the requisite packages with `==`, so we know exactly how the authors originally did all this.

Sample command for `one_shot_pruning.py` in README also includes the command-line arguments `--data-dir` and `--sparseml-recipe`, when the arguments actually added to argparse are `--data_dir` and `--sparseml_recipe`, and arguments `-b` and `--experiment` that don't seem to exist.

**Limitations:**

Cite computation cost (at training/pruning, not inference/deployment time) as the limitation. The Fischer matrix will still grow quadratically in the number of parameters. This is unfortunate given the likelihood that the largest models that would benefit the most from pruning, in absolute terms. (And it is an oft-observed principle in ML that simple methods that scale better are usually more impactful.)

---

> ### Author Rebuttal · Authors · 2023-08-09
>
> Thank you for your feedback, which we address in detail below.
>
> > iv) Largely superseded by Optimal BERT Surgeon in a lot of settings
>
> In short, we emphasize that the scalability of both methods you reference (Optimal BERT Surgeon (OBS) and CAP) is essentially the same, while CAP is significantly more accurate across all the settings we tried.
>
> Both OBS and CAP use the same 2nd order approximation (the block-wise empirical Fisher). The computational and storage cost of both OBS and CAP are dominated by the estimate of the Fisher inverse. Specifically, most of the practical runtime is taken by the forward-backward passes required for estimating the Empirical Fisher at a pruning step. Then, the Optimal BERT Surgeon selects a large group of weights at a single step and prunes it. By contrast, our algorithm eliminates weights one-by-one and accounts for the change of correlations and importance of a particular weight after each previous weight.
>
> The key theoretical contribution of our work is a new algorithm for efficiently resolving these inter-weight correlations which arise during pruning: essentially, CAP allows us to efficiently “simulate” OBS weight removal one-weight-at-a-time (which would take massive time to run iteratively). In turn, taking correlations into account leads to much more accurate pruning, as we illustrated in Figures 2 and 3. If we remove this correlation-solving component, our approach becomes identical to OBS.
>
> In practice, the runtime and scalability cost of the CAP component is negligible relative to the cost of obtaining the Fisher estimate. We have mentioned this in the main text (l.352), and detailed it in Appendix G: our method has essentially the same runtime and memory costs compared to WoodFisher/Optimal BERT Surgeon, while being significantly more accurate.
>
> On methodology, we would like to emphasize that we have considered the best available implementation of the Optimal Brain Surgeon Framework, which is the Optimal BERT Surgeon. The hyperparameter tuning procedure was the same for all methods--we tuned each method independently, and always selected the best hyperparameter configurations for each method in turn. Details on this are provided in the Appendix. Hence, we believe that our comparisons are fair.
>
> > v) Source of improvement in end metrics remains unclear
>
> As noted above, our contribution is an efficient exact weight correlation solver during pruning: if we remove the correlation solver from CAP, we obtain an instance of OBS using a block-wise Fisher approximation, which is essentially WoodFisher/Optimal BERT Surgeon.
> Thus, our one-shot pruning comparisons essentially provide one of the key ablation study that the reviewer is asking for: WoodFisher/OBS is CAP without the correlation solving. So, in the comparisons with WoodFisher/OBS, we are ablating the key component of our method. We have provided this comparison, showing the impact of correlation solving, in Figure 2 and Table 4 in the main text, and Figure 6, Table 6, Figure 7, Table 7, and Figure 8 in the Appendix, across several models and tasks.
>
> Further, we have shown that correlation solving (our method) has a significant effect, beyond just accuracy: in Figure 3 (main text), we illustrated that the weight configurations chosen by CAP and WoodFisher/OBS are significantly different: the sparse weights obtained after one-shot pruning land at different points in the loss basin.
>
> In Appendix Figure 7, we show exactly the same one-shot effects for a ResNet50-D model, whereas Table 6 presents results for a ConvNext model. This validates the fact that our results hold across very different architectures. Moreover, we have also shown results on different tasks (DeTR).
>
> Appendix C performs ablations for the fine-tuning parameters (learning rate schedule and augmentation procedure), validating our choice of schedule. The one-shot + fine-tuning results in Appendix Tables 6 and 7 confirm the effect that the reviewer is referring to: fine-tuning usually reduces the gap between methods, but the differences remain significant, especially at large sparsities. (E.g., CAP produces a 90%-sparse model that is more accurate by 5 Top-1 points relative to WF, on ConvNext-Small.)
>
> In sum, we believe that all these results support our claim that correlation solving can have a consistent significant impact on pruning accuracy, across model scales, model types, and tasks.
>
> (For reference, WoodFisher and Optimal BERT Surgeon implement the same empirical Fisher approximation, but Optimal BERT Surgeon presents a more efficient implementation, with parameters specifically chosen for accuracy and scalability on Transformer models.)
>
> > Code: Some miscellaneous minor comment from trying the included code
>
> We would like to sincerely thank you for your effort in trying out our code, and we apologize for the dependency issues you encountered. We have taken all your suggestions into account to produce an improved version of the code package.
>
> > Limitations
>
> Indeed, computational cost is an issue for all approximate second-order methods in deep learning, which we see as an intriguing challenge to address. Please note, however, that the blockwise Fisher matrix approximation used in our work doesn’t scale quadratically with the dimension: it is linear in the dimension times block size: the algorithm requires $O(d B)$ memory and has $O(d B^2)$ runtime, as mentioned in Section 3.1.  Larger models benefit more from pruning, but compression of moderate-size models (e.g. with 1-50M parameters) is still of great practical use for inference on edge devices, constrained in their compute power. Our work shows significant practical speed-ups for unstructured sparsity on commodity CPUs, therefore it can have real-world impact.
>
> To further address this concern of scalability, in the response to Reviewer JVcP and general response we present a much more efficient approximate version of CAP (FastCAP).

---

> > ### Comment · Reviewer_kMpj · 2023-08-21
> >
> > >  Thus, our one-shot pruning comparisons essentially provide one of the key ablation study that the reviewer is asking for: WoodFisher/OBS is CAP without the correlation solving
> >
> > This seems correct, and a good experiment without confounds (since its comparing under the simple & shared one-shot procedure) in the case of Figure 2 & similar experiments in the supplement. Though I'm perhaps looking in the wrong place for Table 4 in the main text?
> >
> > > general response we present a much more efficient approximate version of CAP (FastCAP).
> >
> > Where ought we look for more details on FastCAP? The description given in the top-level comment seems elides a lot that might be needed for a full review.
> >
> > > compression of moderate-size models (e.g. with 1-50M parameters) is still of great practical use for inference on edge devices
> >
> > Definitely true! Perhaps this is scalable enough.

---

> > > ### Author Response · Authors · 2023-08-21
> > >
> > > Thank you for your response!
> > > Please see replies inline:
> > >
> > > > This seems correct, and a good experiment without confounds (since its comparing under the simple & shared one-shot procedure) in the case of Figure 2 & similar experiments in the supplement. Though I'm perhaps looking in the wrong place for Table 4 in the main text?
> > >
> > > We are very glad that the reviewer found this part of the response clarifying. We will also highlight this point in the next revision.
> > >
> > > We apologize for the confusing table reference: we meant Table 1, not Table 4. More precisely, Table 1 shows the comparison of one-shot pruning performance of various methods on large (CLIP-sized) models.
> > >
> > >
> > > >  Where ought we look for more details on FastCAP? The description given in the top-level comment seems elides a lot that might be needed for a full review.
> > >
> > > To address this, we provide the full pseudocode for FastCAP below; for simplicity, we provide the code for a single weight. The same procedure is applied for all weights. The main two components of the CAP/FastCAP algorithm are:
> > >
> > > 1. **Empirical Fisher estimate.** The original CAP has specific Fisher block for each output channel, whereas the FastCAP version averages blocks across the output channel dimension. (This reduces storage complexity.)
> > >
> > > 2. **Weight elimination and update iteration.** If the original CAP eliminates weights in greedy optimal order, FastCAP prunes weights in a fixed order. (This reduces computational complexity.)
> > >
> > > Notation:
> > > * `B` - Fisher block size
> > > * `b` - block size in iterative pruning algorithm
> > > * `N` - number of gradients
> > > * `N_B` - number of blocks per parameter
> > > 1) **Fisher accumulation**
> > >
> > > `F` - matrix with zeros with shape `(B, B)`
> > >
> > > **for** i=1:N **do**
> > >
> > > &ensp; reshape `dL_i / dw` to `(N_B, B)`
> > >
> > > &ensp; `F +=  (1 / N) (dL_i / dw)^T dL_i / dw `  // average Fisher blocks across the layer output dimension
> > >
> > >
> > > **end for**
> > >
> > > 2) **Iterative pruning**
> > >
> > > `M  ← 1_{N_B x B}` // binary pruning mask
> > >
> > > `E ← 0_{N_B x b}` // block pruning errors
> > >
> > > `F ← Cholesky({F}^{-1})`  // efficient form of the Fisher inverse
> > >
> > > **for** `i`=`0`, `b`, `2b`, … **do**
> > >
> > > &ensp; // select elements to prune
> > >
> > > &ensp; `M_{:,j:(j+b)}` ← mask of  `(1 − p)`% weights `w_k`  `in W_{:,j:(j+b)}`  with largest `w_k^2 / F^{-1}]_{kk}^2`
> > >
> > > &ensp; for j=`i`, …, `i+b-1` **do**
> > >
> > > &ensp; &ensp; `E_{:,j−i} ← W_{:,j} /  [F^{-1}]_{jj}` // pruning error
> > >
> > > &ensp; &ensp; `E_{:,j−i} ← (1- M_{:, j}) E_{:,j−i}`  // freeze weights
> > >
> > > &ensp; &ensp; `W_{:,j:(i+b)}  ← W_{:,j:(i+b)} - E_{:,j−i} F_{j,j:(i+B)^{-1}} ` // update remaining weights to correct output
> > >
> > > &ensp; **end for**
> > >
> > > **end for**
> > >
> > > `W ← W * M` // set pruned weights to 0
> > >
> > >
> > > As mentioned in the earlier response, the Fisher is the same for all blocks, reducing space complexity to $O(B^2)$. We adapted the trick of maintaining the Fisher inverse in Cholesky form from the SparseGPT paper, to account for iterative elimination of the weights in a fixed order. This allows running the iterative process efficiently.

---

### Official Review · Reviewer_JVcP · 2023-07-05

**Soundness:** 4 excellent
**Presentation:** 4 excellent
**Contribution:** 3 good
**Rating:** 7
**Confidence:** 4

**Summary:**

This paper proposes to consider the correlation between pruned elements in pruning deep neural netowkr models. The paper provides an efficient algorithm to distangle the correlation into a sparse regression problem, and propose a fast solver to find the solution. Further exploration is performed on the learning rate scheduling and data augmentation of the pruning and finetuning process. The final method results in better model size-accuracy tradeoff comparing to previous method with less training cost.

**Strengths:**

1. This paper provides well motivated method on model pruning with solid theortical justification
2. The proposed reformulation of sparse minimization and the fast solver of the optimization is novel an have significant impact on model compression
3. Adequate experiments are provided to show the effectiveness of the proposed method, including pruning advanced highly accurate models
4. The paper is overall well written and easy to follow.

**Weaknesses:**

As discussed in the limitation, the cost of considering full correlation is still prohibitive on large models. It would be interesting to see some relaxation of the proposed method that can have less cost while maintaining most of the good performance. More discussion is encouraged on the potential relaxation strategy, and the tradeoff between optimization cost and performance.



**Questions:**

See weakness.

**Limitations:**

The limitation and social impacts have been adequately discussed.

---

> ### Author Rebuttal · Authors · 2023-08-09
>
> Thank you for your very insightful feedback!
>
> Regarding scaling, currently, CAP can scale easily to models with hundreds of millions of parameters, with reasonable block sizes and reasonable runtime (< 30 minutes / pruning step on 1 GPU). Yet, you are right in that it would be challenging to scale CAP to billion-parameter models, because of the necessity of storing and processing blocks of the Fisher inverse.
> Motivated in part by your suggestion, we observed that there is a relaxation of CAP which can easily scale to massive models, via two steps:
>
> * First, one can average Fisher blocks across the layer output dimension and use these blocks as proxy of the Hessian, where each Hessian matrix is the same for all output channels. (This step is similar to the Layer-Wise Optimal Surgeon paper [1].)
> * Secondly, following the recent SparseGPT paper [2], instead of eliminating the weights in the greedy-optimal order, i.e. always selecting the weight whose elimination leads to the smallest increase of loss, one can prune them in some fixed order, fixed across all output dimensions. Specifically, for this we adapted the iterative removal process from SparseGPT, which is more GPU-friendly and scales better with increase of model size.
>
> The resulting modification is an approximation of CAP: the main constituents - block-wise Fisher approximation and resolution of inter-weight correlations are the same as in the original algorithm. The two approximations above lead to reduction of the memory footprint and the runtime which are linear in the embedding dimension: in practice, this is 100x-1000x compared to the original CAP. Thus, FastCAP is easily scalable to billion-parameter models with reasonable computational cost.
>
> To validate performance of FastCAP we applied it for pruning of large ViT models in a one-shot setting, specifically `eva_giant_patch14_224.clip_ft_in1k` from the `timm` library. This model has **>1B** parameters. We compared the top-1 accuracy on ImageNet of FastCAP vs Magnitude pruner. One can see that FastCAP significantly outperforms the baseline. At the same time, FastCAP has a reasonable runtime and memory footprint at this scale. Pruning step takes ~400 seconds on a single A100 GPU.
>
> Table 1. Performance of large ViT after 1-shot pruning.
> |   Model   | Sparsity |   Method  | Accuracy (%) |
> |:---------:|:--------:|:---------:|:------------:|
> | EVA ViT-G |     0    |     -     |     88.7     |
> |           |    50    | Magnitude |     87.9     |
> |           |          |  FastCAP  |     88.1     |
> |           |    60    | Magnitude |     85.5     |
> |           |          |  FastCAP  |     86.3     |
> |           |    70    | Magnitude |     64.3     |
> |           |          |  FastCAP  |     76.1     |
>
> [1] Dong, Xin, Shangyu Chen, and Sinno Pan. "Learning to prune deep neural networks via layer-wise optimal brain surgeon." Advances in neural information processing systems 30 (2017).
>
> [2] Frantar, Elias, and Dan Alistarh. "Massive language models can be accurately pruned in one-shot." arXiv preprint arXiv:2301.00774 (2023).

---

> > ### Comment · Reviewer_JVcP · 2023-08-21
> >
> > I would like to thank the author for the response. I'm satisfied with the response and will keep my score.

---

### Official Review · Reviewer_wReA · 2023-07-09

**Soundness:** 3 good
**Presentation:** 4 excellent
**Contribution:** 3 good
**Rating:** 5
**Confidence:** 4

**Summary:**

The paper proposed a Correlation Aware Pruner (CAP) , a new unstructured pruning framework capable to prune models to high sparsity. It takes into account weight correlations. To do this, the paper reformulate the OBS multi-weight pruning problem: when using the empirical Fisher approximation, the problem of finding the optimal set of weights to be removed, while taking correlations into account, is equivalent to the problem of finding the set of sparse weights which best preserve the original correlation between the dense weights and the gradients on an fixed set of samples.

On top of that, the paper also applies a series of training techniques to improve training: Learning Rate Schedule,  Regularization and Augmentation, Efficient Sparsity Sweeps. Results show better performance compared to existing works, expecially at high compression ratio

**Strengths:**

The paper is well written, and has good logic. Great amount of experiments and ablations discussion to strengthen the proposed method.

**Weaknesses:**

Unstructured pruning is hard to have actual hardware speedup due to the irregular sparsity. Although the proposed method can maintain high accuracy at extreme prune ratios, it may just be just theoretically more efficient.

The author is encouraged to add inference latency or throughput in the results tables 1 & 2

**Questions:**

See weekness

**Limitations:**

Please address the limitations and potential negative societal impact in the revision.

---

> ### Author Rebuttal · Authors · 2023-08-09
>
> Thank you for your review and comments!
>
> It is true that, traditionally, sparsity is harder to leverage for computational speedups. However, unstructured sparsity is now supported with speedups on CPU, and 2:4 semi-structured sparsity is supported with speedups on NVIDIA GPUs. Our method can create accurate models targeting both sparsity types:
>
> * Figure 4 shows end-to-end throughput speedups of more than 2x for CPU deployments when running CAP models using the DeepSparse inference engine (supporting unstructured sparsity).
> * In Appendix H.3, we provide accuracy results for several models with 2:4 semi-structured sparsity, with minor or no performance drop compared to the original dense model. This format is natively supported with speedups by all modern NVIDIA GPUs (Ampere, Hopper) [1].
>
> Following your suggestion, we will add the corresponding inference throughput speedups in Tables 1 and 2. The numbers can currently be read from Figure 4, which provides the desired data in speedup-vs-accuracy format. Please note that, e.g. for DeiT-Tiny, we obtain a real-world speedup of > 2x at negligible accuracy loss.
>
> The main limitation of our method is the scalability, since the memory footprint and runtime becomes prohibitively expensive for models with 1B parameters and more. However, one can propose a relaxed version with some additional approximations that can be scaled to large models. We describe it in more detail in general response.
>
> Sparse models are not expected to show more malicious behavior compared to the dense models. However, as any other kind of technology, they can be applied both for good and bad purposes. The main potential outcome of our work is the speedup and more widespread adoption of modern computer vision architectures in resource constrained setup, such as inference on edge devices.
>
> [1] https://developer.nvidia.com/blog/exploiting-ampere-structured-sparsity-with-cusparselt/

---

> > ### Comment · Reviewer_wReA · 2023-08-21
> >
> > Thank you for your clarifications. I would like to raise my score to week accept (6)

---

> > > ### Author Response · Authors · 2023-08-21
> > > **Thank you!**
> > >
> > > We would like to thank the reviewer for their response, and for raising their score!
> > >
> > > P.S.: As far as we can tell, the score has remained the same in the Author Console.

---

### Author Rebuttal · Authors · 2023-08-09

We thank the reviewers for their feedback and comments on our work.

Below is the summary of the main concerns and questions addressed in our rebuttal:

**1. Difference between WoodFisher (Optimal Brain Surgeon) and CAP.**

In the response, we emphasized that our contribution is not another approximation of Fisher matrix, but a new algorithm resolving the correlations between the weights during the pruning process. Specifically, CAP allows the user to efficiently emulate the ideal Optimal Brain Surgeon (OBS) process of pruning one-weight-at-a-time, adjusting the remaining weights after each removal. As such, it is the first implementation of “true” OBS at scale. If we completely remove correlation solving from CAP and perform one-shot pruning, we simply obtain a vanilla instance of OBS, essentially WoodFisher.

Thus, our comparisons with regular instances of OBS such as WoodFisher/Optimal BERT Surgeon, which do not solve for correlations, essentially showcase the power of correlation solving. These results are shown in Figures 2 and 3 and Table 4 in the main text and  Figure 6, Table 6, Figure 7, Table 7, and Figure 8 in the Appendix, show that resolving correlations accurately can have a major impact on the accuracy of the pruned model. Appendix Figure 12 performs a very fine-grained analysis of the impact of correlation solving on the accuracy of pruned models.

**2. CAP Scalability.**

The version of CAP we presented in our submission can scale to models with < 1B  parameters, covering most standard vision models. Its key computational and memory cost comes because of computing and maintaining the inverse Fisher approximation. This is the same as other methods, such as WoodFisher/Optimal BERT Surgeon, and the running time of CAP is essentially the same as these prior methods, while being significantly more accurate.

To address this scalability concern in full, in response to Reviewer jVCP we present a much more scalable approximate variant of CAP, called FastCAP, which is inspired by the recent SparseGPT scalable pruner (arXiv::2301.00774). Specifically, FastCAP reduces compute and memory cost by ~100x by leveraging a cheaper Fisher approximation and relaxing the optimal greedy order of pruning. The other method components stay the same. FastCAP can be scaled to billion-parameter vision models, i.e. we can prune ViT-Giant in a few minutes, while achieving reasonable performance in one shot-setting for 50-70% sparsity. We present a simple description of FastCAP in the responses to Reviewers jVCP and  kMpj, and plan to provide the full results in the next revision.  The results for pruning of large ViT model are presented in **Table 1**.

Table 1. Performance of 1-shot pruning on ViT-Giant.
|   Model   | Sparsity |   Method  | Accuracy (%) |
|:---------:|:--------:|:---------:|:------------:|
| EVA ViT-G |     0    |     -     |     88.7     |
|           |    50    | Magnitude |     87.9     |
|           |          |  FastCAP  |     88.1     |
|           |    60    | Magnitude |     85.5     |
|           |          |  FastCAP  |     86.3     |
|           |    70    | Magnitude |     64.3     |
|           |          |  FastCAP  |     76.1     |

**3. Practical Speedups.**

We presented end-to-end speedup results on CPUs in the original submission: for instance, we obtain speedups of 1.5--2x across ViT family models, with negligible accuracy impact.  Although general sparsity is not supported on GPU hardware, 2:4 structured sparsity is natively supported in modern (Ampere and Hopper) GPU architectures, and leads to speedups of the tensor multiplication operations. Our method can be applied to create 2:4 sparse models as well, with a minor modification to the sparsity mask constraint.

Accuracy results for the resulting 2:4-sparse models were presented in Appendix H.3. For convenience, we present them in **Table 2** below. The resulting models could be executed with speedups via the NVIDIA TensorRT GPU inference engine, at relatively minor accuracy drops (0.8-0.1%) relative to the dense baselines.

Table 2. Semi-structured 2:4 pruning of ViT models followed by 10 epochs of finetuning

|    Model    | Method | Top1-Accuracy (\%) |
|:-----------:|:------:|:------------------:|
|  DeiT-Tiny  |    -   |        72.2        |
|             |   GM   |        68.8        |
|             |   CAP  |        71.5        |
|  DeiT-Small |    -   |         79.8       |
|             |    GM  |         77.9       |
|             |   CAP  |         79.0       |
|   DeiT-Base |    -   |         81.8       |
|             |    GM  |         81.2       |
|             |   CAP  |         81.3       |

We look forward to an engaging discussion!

---

### Author Response · Authors · 2023-08-19
**Discussion reminder**

Dear Reviewers,

Given that the discussion period is ending shortly and we have received no feedback so far, we wanted to send a gentle reminder regarding our review responses.

We would really appreciate your feedback on our rebuttal, especially regarding the added experimental results and clarifications regarding our method.

With best regards,

The CAP Authors

---

### Decision · Program_Chairs · 2023-09-21

**Decision:**

Accept (poster)

**Comment:**

Summary
This paper presents an approach to pruning models by looking at the correlations in the model weights. The authors reduce this problem to solving a sparse regression problem that allows them to achieve a high degree of sparsity in the final pruned model. The paper provides a good connection to theory for motivating this way of pruning and achieves a good tradeoff in performance and model weights.

Reviews & Justification
The authors addressed the reviewers concerns successfully. RwReA increased their rating to WA (didn’t mark); RkMpj’s concerns were addressed but didn’t update their rating. Overall, this paper makes a valuable addition to the model pruning.